# A critical role of action-related functional networks in Gilles de la Tourette syndrome

Juan Carlos Baldermann [1,2] ✉, Jan Niklas Petry-Schmelzer[1], Thomas Schüller [2], Lin Mahfoud[1], Gregor A. Brandt [1], Till A. Dembek[1], Christina van der Linden [1], Joachim K. Krauss [3], Natalia Szejko[4,5,6], Kirsten R. Müller-Vahl [5], Christos Ganos[7], Bassam Al-Fatly [8], Petra Heiden[9], Domenico Servello[10], Tommaso Galbiati [10], Kara A. Johnson [11], Christopher R. Butson[11,12], Michael S. Okun [11], Pablo Andrade[9], Katharina Domschke [1,13], Gereon R. Fink [1,14], Michael D. Fox [15], Andreas Horn [15,16,17,18], Jens Kuhn [19,20], Veerle Visser-Vandewalle [9] & Michael T. Barbe[2]

Gilles de la Tourette Syndrome (GTS) is a chronic tic disorder, characterized by unwanted motor actions and vocalizations. While brain stimulation techniques show promise in reducing tic severity, optimal target networks are not well-defined. Here, we leverage datasets from two independent deep brain stimulation (DBS) cohorts and a cohort of tic-inducing lesions to infer critical networks for treatment and occurrence of tics by mapping stimulation sites and lesions to a functional connectome derived from 1,000 healthy participants. We find that greater tic reduction is linked to higher connectivity of DBS sites (N = 37) with action-related functional resting-state networks, i.e., the cingulo-opercular (r = 0.62; p < 0.001) and somato-cognitive action networks (r = 0.47; p = 0.002). Regions of the cingulo-opercular network best match the optimal connectivity profiles of thalamic DBS. We replicate the significance of targeting cingulo-opercular and somato-cognitive action network connectivity in an independent DBS cohort (N = 10). Finally, we demonstrate that tic-inducing brain lesions (N = 22) exhibit similar connectivity to these networks. Collectively, these results suggest a critical role for these action-related networks in the pathophysiology and treatment of GTS.

Tics are typified by repetitive, sudden actions in the form of unwanted motor behaviour or vocalizations. Gilles de la Tourette syndrome (GTS) is a chronic tic disorder, wherein both motor and vocal tics occur. Tics can become so intrusive that they profoundly impede daily functions. This disruption may stem from interruptions of goal-oriented behaviour, the tics' social inappropriateness, or even the pain and self-harm they potentially induce. Deep brain stimulation (DBS) has repeatedly demonstrated potential in mitigating tic severity in treatment-refractory cases that do not respond sufficiently to pharmacotherapy and behavioural therapies. However, DBS outcomes remain heterogeneous[1,2]. One significant limitation of DBS for GTS has been the insufficient understanding of the optimal brain networks to be targeted.

Among movement disorders, tics have a unique phenomenology. Typically, tics are preceded by an aversive sensory phenomenon, the premonitory urge, that diminishes after tic execution[3]. Consequently, it is debated whether tics are involuntary or if they form a volitional motor response to an unwanted, pathological sensation[4]. The severity of tics fluctuates in response to environmental influences and mental states, e.g., aggravating during stress or mitigating during mindfulness

practices[5]. Psychiatric comorbidities such as obsessive-compulsive disorder (OCD), attention deficit hyperactivity disorder (ADHD) and anxiety are prevalent in approximately 90% of patients[6]. These clinical observations underline that tics cannot be seen as a pure function of the motor system, but that sensory and affective brain states influence them. Indeed, functional magnetic resonance imaging (fMRI) studies show widespread alterations of brain activity in patients with chronic tic disorders. While tic execution has been primarily linked to activity in the pre-and primary motor cortices, activity in sensory and prefrontal areas such as the parietal cortex, insula, dorsal anterior cingulate cortex (dACC), and the dorsomedial prefrontal cortex (DMPFC)/ supplementary motor area (SMA) is associated with experiencing the premonitory urge (see refs. [3,7,8] for respective reviews). These networks activate on both hemispheres before and during tic executions[9,10], although different case-control studies suggest a right-hemispheric functional dominance in GTS compared to healthy controls[11,12].

Conceptualising tic disorders as a multinetwork disorder poses the question of how these multiplex circuitries are intertwined and how they relate to neuromodulatory interventions for GTS. Gordon et al. [13]. recently introduced a novel functional brain network pivotal for action control that links the primary motor cortex with limbic and sensory networks. Employing advanced precision fMRI techniques, they identified regions within the primary motor cortex (M1) that diverged from the established understanding of specific bodily movement representation. Instead, these areas, named inter-effector regions, exhibited activity during action planning and intricate axial body movements. The inter-effector regions were highly interconnected, forming a novel network within M1, termed the somato-cognitive action network (SCAN). Further, the SCAN displayed heightened connectivity with the cingulo-opercular network (CON)—a system involved in arousal, error detection and pain sensation. Very recently, the authors who first described CON based on their anatomy, argued that this network can be functionally characterised as an action-mode network[14]. This term assembles the different CON-associated functions associated with mental processes that lead to action (in the following, we will continue using the term CON to ensure congruency with most of the literature). The CON encompasses sensory and prefrontal areas in the insula/operculum, dACC, DMPF up to the SMA, supramarginal gyrus and the anterior prefrontal cortex[15–17].

Notably, it has previously been shown that patients with chronic tic disorders show altered activity of the CON[18,19]. Further, critical areas of the CON, i.e., the insula, the dACC, SMA and the supramarginal gyrus, are closely linked to the processing of tics and the preceding premonitory urge[3,9–11,20–22]. This novel insight into the intricate functional interactions of the motor system with prefrontal and sensory networks offers potential therapeutic implications for neuropsychiatric disorders characterised by motor symptomatology, such as tic disorders. Gordon et al. [13]. further found that the SCAN exhibits robust connectivity with intralaminar thalamic nuclei, especially the centromedian nucleus (CM)—the most common DBS target for tic disorders[1,2]. While different studies aimed to characterise the structural networks underlying DBS, functional networks to be targeted are less studied. In previous data-driven attempts in a small cohort ($N = 15$) of patients treated with thalamic DBS, we identified sensorimotor networks[23] and the insula[24] to be associated with tic reduction after DBS. However, this analysis did not specifically investigate known large-scale functional networks related to action, i.e., CON and SCAN, and was limited by their restricted sample size.

Given the SCAN's assumed role in action planning, its prominent connectivity with current treatment targets for tics and its close relationship with the urge-related CON, we theorized the importance of targeting these networks for tic relief. Here, we investigated this relationship empirically: we hypothesised that functional connectivity of individual stimulation sites with the SCAN and CON could explain outcome variance in a multicenter cohort of patients with tic disorders treated with thalamic DBS (Fig. 1). We then aimed to replicate our observations in an independent sample of patients with tic disorders treated with thalamic DBS. Finally, we tested whether tic-inducing brain lesions exhibit similar connectivity to the SCAN and CON.

## Results
### Clinical data, locations of stimulation sites and averaged connectivity
In the main analysis, we included 37 patients with severe GTS (mean age: 32.4 years ± 10.8 years (standard deviation); range: 18–65; 5 females). 19/37 patients suffered from comorbid obsessive-compulsive behaviour (OCB). Six to twelve months after continuous DBS of the thalamus, the average tic severity improved by 32.5% (± 21.3). There was a weak but non-significant correlation between age and tic reduction ($r = -2.84$; $p = 0.088$), and no significant differences could be discerned between outcomes of males and females ($z = 0.845$; $p = 0.423$) or outcomes of patients with and without OCB ($z = 0.258$; $p = 0.796$).

Electrode localization within the medial and motor thalamus was confirmed for all patients (Fig. 2a). Computation of stimulation sites confirmed that the CM and VOI, i.e., the two main target nuclei across the sample, were stimulated in 85% and 76% of patients, respectively. Other nuclei that could potentially mediate tic reduction, such as the VA nucleus or the nucleus ventro-oralis posterior, were stimulated in 50% and 67% of the patients, respectively (Fig. 2a). Paired t-testing between averaged MNI coordinates of each electrode showed no significant differences in electrode location between the right and left hemispheres on a group level for coordinates x ($t = -0.74$; $p = 0.589$), y ($t = -1.0$; $p = 0.405$) and z ($t = -0.97$; $p = 0.43$). On average, DBS sites demonstrated functional connectivity to areas closely matching the CON. When qualitatively assessing connectivity values within M1, DBS sites exhibited positive functional connectivity to the inter-effector regions of the SCAN, while the somatotopic M1 was negatively connected (Fig. 2b).

### Functional networks related to tic reduction
To test whether connectivity to the CON and SCAN was associated with treatment outcomes, we performed an a priori-defined ROI analysis by averaging all connectivity estimates of stimulation sites in all voxels within the respective regions for each individual. Connectivity of the bilateral stimulation sites with the CON was significantly correlated with tic reduction ($r = 0.62$; $p < 0.001$). Further, connectivity with the SCAN also correlated significantly with tic reduction ($r = 0.47$; $p = 0.002$) (Fig. 3a). Notably, while connectivity to the CON was positive for all patients, connectivity to the SCAN was negative for 58 % of non-responders (i.e., tic reduction < 25 %[25]) and positive for 85 % of full-responders responders (i.e., tic reduction ≥ 35 %). To determine the specificity of these associations, we performed post-hoc correlations between tic reduction and other large-scale functional resting-state networks. Only CON and SCAN showed a significant positive association with tic reduction ($p_{FDR} < 0.05$). Connectivity with the sensorimotor ($p_{FDR} = 0.063$), central executive ($p_{FDR} = 0.096$) and visual ($p_{FDR} = 0.096$) networks did not show a significant correlation while connectivity with the language ($p_{FDR} = 0.047$) and default mode networks ($p_{FDR} = 0.0035$) was significantly anticorrelated with tic reduction (Fig. 3b). A post-hoc analysis revealed that the positive association of tic reduction with connectivity to the SCAN was mainly driven by the inferior and middle inter-effector regions, both showing strong significant correlations ($r = 0.596$; $p < 0.001$ and $r = 0.513$; $p < 0.001$). In contrast, the superior inter-effector regions showed a weak correlation that approaches but does not reach statistical significance ($r = 0.276$; $p = 0.052$) (Fig. 3c). Since the sensorimotor network showed a trend towards significance, we performed a post-hoc analysis to investigate which parts of this network drove the effect. Both the precentral

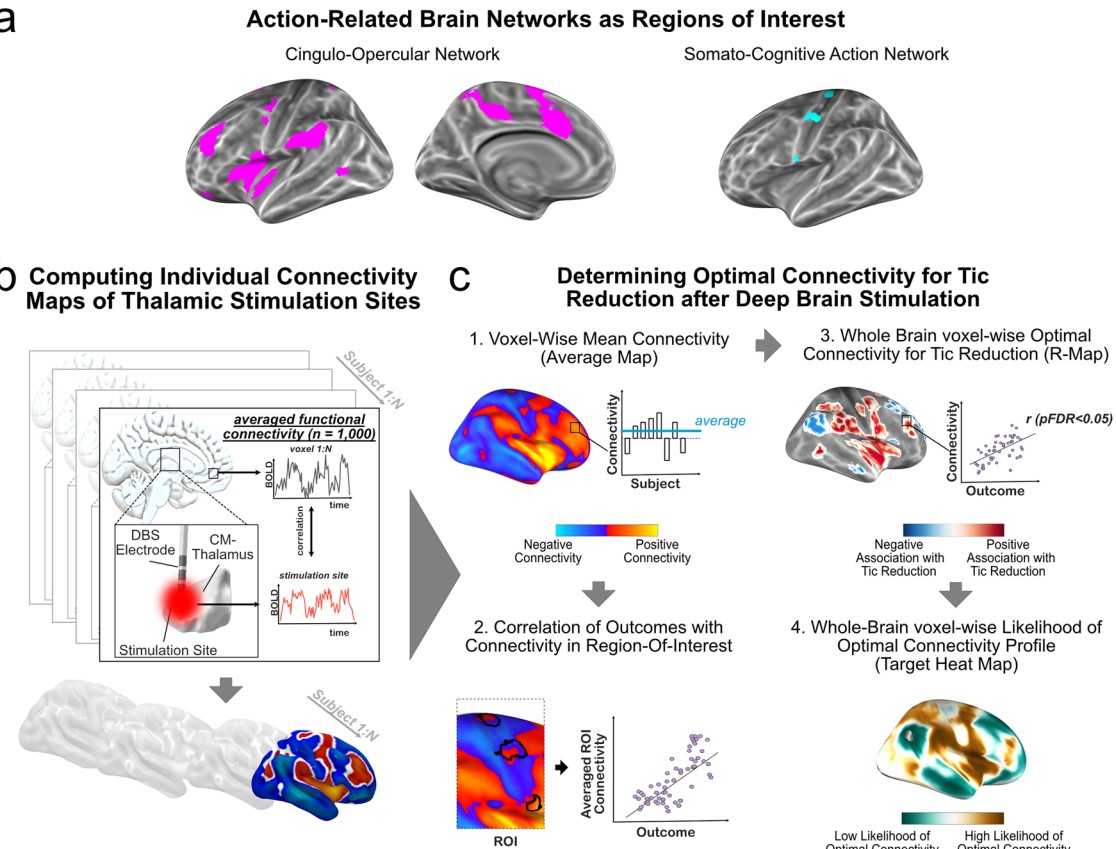

**Fig. 1 | Method Overview. a** As regions of interest, we chose functional networks based on resting-state connectivity known to be involved during actions in humans. The cingulo-opercular network (CON), also referred to as the action-mode network, is linked to the processing of arousal, error detection and pain sensation and thus states that call for action. The somato-cognitive action network (SCAN) has recently been described as a network relevant to action planning and complex body movements. Both networks show high connectivity with each other. **b** For each subject in a cohort of patients with GTS undergoing thalamic DBS for tic reduction ($N = 37$), electrodes and the volume of activated tissue (i.e., stimulation site) were reconstructed in standard space. Using a publicly available resting-state functional connectome acquired in healthy participants ($N = 1000$), we computed the functional connectivity of each bilateral pair of stimulation sites with all other brain voxels. The resulting connectivity maps were then used for group analysis. **c** Initially (1), a map of voxel-wise average connectivity was computed (termed average-map). Subsequently (2), the average connectivity in priori-defined regions of interest, i.e., CON and SCAN, was correlated with tic reduction post-DBS. In an alternative data-driven whole-brain approach (3), we calculated a map of voxel-wise significant correlations (pFDR < 0.05) between connectivity and tic reduction (termed R-map), signifying an optimal connectivity pattern. Lastly (4), the weighted connectivity of this R-map, limited to cortical voxels, to each other brain voxel was computed. This resulted in a map where each voxel contained the likelihood of matching the optimal connectivity profile, suggesting potential novel cortical target networks (termed target heat map).

($r = 0.44$, p_uncorrected = 0.003) and postcentral gyri ($r = 0.34$, p_uncorrected = 0.023) showed positive correlations with outcomes. However, unlike the positive correlations with the CON and SCAN, all patients showed negative connectivity values for the postcentral gyrus, and 92% showed negative connectivity values for the precentral gyrus (Supplementary Fig. 3). Thus, the association of tic reduction with connectivity to the sensorimotor network likely reflects a different mechanism, where reduced or no anticorrelation with the network correlates with better outcomes after DBS. Specifically, this analysis indicates that stimulation sites of top-responders would likely exhibit strong connectivity with the CON and SCAN while showing no connectivity or only low negative connectivity with the SMN.

We then derived a data-driven, whole-brain pattern of functional connectivity significantly associated with tic reduction after DBS by computing voxel-wise correlations between treatment outcomes and connectivity values which were then corrected for multiple comparisons. In this map, termed R-map, a positive association between connectivity and tic reduction was observed for clusters in the insula, the lateral anterior prefrontal cortex and medial prefrontal cortex, including the pars marginalis of the cingulate cortex, dACC, and SMA, the supramarginal gyrus, the premotor cortex and single clusters

within M1 (Fig. 3b). The insular cluster encompassed the posterior short and long gyrus, as well as the middle and anterior short gyrus and the rolandic operculum. Subcortically, the insula cluster extended to posterior parts of the claustrum and encompassed the posterior tip of the putamen (Fig. 3e). On a cortical level, the connectivity pattern linked to greater tic reduction closely matched the CON and the middle and inferior inter-effector regions of the SCAN, matching the ROI analysis. Clusters with a negative association were observed for the precuneus, a cluster surrounding the frontal pole and ventromedial prefrontal cortex and a cluster in the gyrus angularis, reaching to the occipital lobe. Again, the observed pattern was highly symmetric between hemispheres. Quantitatively, the R-map associated with greater tic reduction showed the highest overlap and highest balanced accuracy with the CON (overlap 46% and balanced accuracy 0.89), followed by the SCAN (26% and 0.73). All other networks showed lower overlaps and balanced accuracies (sensorimotor 13% and 0.60; visual 7% and 0.55; central executive 6% and 0.57; language <1% and 0.50; default mode <1 % and 0.48).

As a concluding step, we aimed to obtain a whole-brain map that elucidates potential target networks for tic disorders. To this end, we computed a target heat map by calculating each voxel's weighted

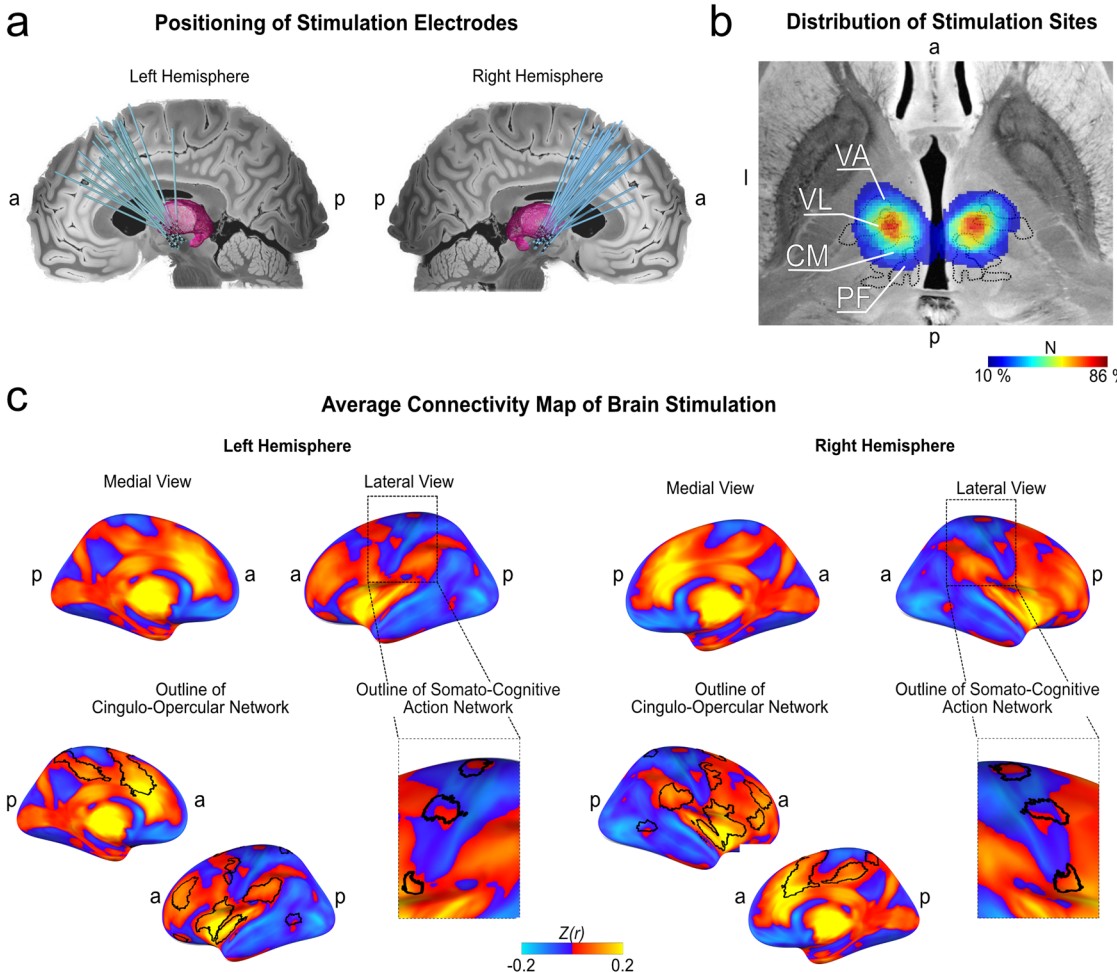

**Fig. 2 | Location of electrodes and stimulation sites with average map of connectivity. a** Reconstruction of electrodes in standard space is shown in the upper and middle figures (the thalamus is highlighted in purple). **b** The distribution of stimulation sites is displayed (MNI coordinate: $z = -2$), with most sites situated at the intersection of the ventral lateral (VL) and centromedian nucleus (CM), further encompassing the ventral anterior nucleus (VA) and the parafascicular complex (PF). The colour bar indicates the percentage of all stimulation sites per hemisphere ($N = 37$) covering the respective voxel. **c** on average, stimulation sites showed heightened connectivity to prefrontal, insular and parietal areas that closely aligned with the cingulo-opercular network (CON), further engaging the medial frontal and cingulate cortex. Within the primary motor cortex (M1), the inter-effector regions of the somato-cognitive-action network (SCAN) were positively connected, while the rest of M1, i.e., the effector regions, were anti-correlated. a = anterior; p = posterior; l = left; r = right.

connectivity with the previously calculated optimal connectivity profile as proposed before[26–28]. The resulting map thus represents the likelihood of matching the cortical connectivity profile of optimal thalamic stimulation. This target heat map further aligned with the CON and distinctively outlined three bridging patterns within M1, accurately reflecting the SCAN (Fig. 4). Within this map, peak clusters (containing voxels with $r > 0.6$) were located in the bilateral supramarginal gyrus, rolandic operculum/anterior dorsal insula, and the SMA, all located within the CON (Supplementary Table 1).

### External replication

We then sought to replicate our results in an independent sample. To this end, we performed the same analysis using the GTS-DBS-Registry sample ($N = 10$). Again, we observed strong associations between tic reduction and connectivity to the CON ($r = 0.54$; $p = 0.047$) and SCAN ($r = 0.83$; $p = 0.003$) (Fig. 4). In addition, computing a similar target heat map in this sample resulted in a similar pattern, again aligning with both CON and SCAN (Fig. 5a). Lastly, we tested whether these networks could be observed in clinical datasets beyond DBS. To this end, we reinvestigated a previously published and unmodified lesion network map of 22 brain lesions that led to the occurrence of tics[24]. In the unthresholded map, we found that the CON showed the highest

number of lesions connected. Within the primary motor cortex, the SCAN was again discernible. Comparing the average number of lesions connected to each voxel within common large-scale functional networks confirmed that most lesions were connected to the CON (69.4 % ± 16.1), followed by the SCAN (43.0 % ± 14.5). Sensorimotor (36.8 % ± 23.0), central executive (36.3 % ± 15.1), language (29.0 % ± 20.8), default mode (22.5 % ± 14.6) and visual network (12.9 % ± 6.7) all showed significantly less connected lesions across voxels than both CON and SCAN (all $p < 0.001$) (Fig. 5b).

### Discussion

By mapping the normative functional connectivity profile of a multi-center cohort of patients with GTS undergoing thalamic DBS and tic-inducing lesions, we investigated the role of action-related functional networks in tic disorders. We specifically hypothesised that functional connectivity to action-related networks, namely the CON and the recently identified SCAN, would be associated with tic reduction. Our study confirmed that thalamic DBS with higher connectivity to both networks resulted in more robust tic reduction. Furthermore, the connectivity of regions within the CON and SCAN matched with the connectivity of optimal thalamic DBS. Finally, we could replicate the importance of the CON and SCAN in an independent DBS dataset and

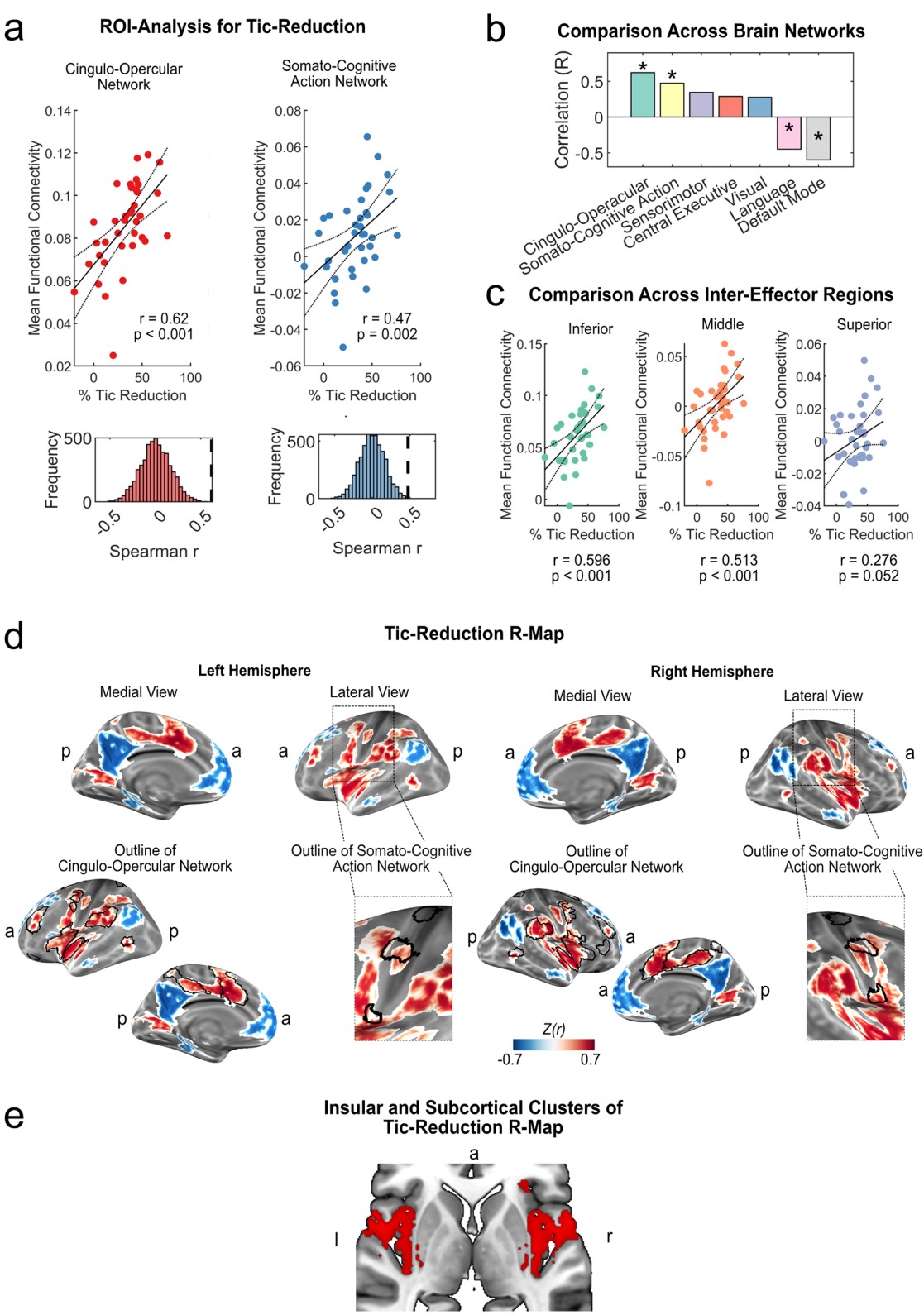

**a** ROI-Analysis for Tic-Reduction

**b** Comparison Across Brain Networks

**c** Comparison Across Inter-Effector Regions

**d** Tic-Reduction R-Map

**e** Insular and Subcortical Clusters of Tic-Reduction R-Map

in a map derived from brain lesions resulting in tic occurrence, further suggesting an involvement of these networks in both the pathophysiology and treatment of tic disorders.

DBS is a highly focal intervention targeting brain areas as small as a few millimetres. Beyond the targeted area, DBS affects long-range pathways and polysynaptic networks[29]. Investigating optimal network targets, rather than isolated optimal spots within a target region, offers

two main advantages. Firstly, it provides insights into the mechanism underlying effective neuromodulation. Secondly, this network approach can elucidate other critical areas potentially relevant for effective neuromodulation. Our analysis demonstrates the application of this approach in the context of tic disorders such as GTS in a comparatively large and multicentered cohort. Functionally, the connectivity of two multifocal cortical brain networks explained treatment

**Fig. 3 | Association of DBS connectivity and tic reduction. a** In a region-of-interest (ROI) analysis, we correlated the percentage tic reduction after DBS with the respective connectivity of stimulation sites ($N = 37$) with the cingulo-opercular (CON) (top left) and somato-cognitive action network (SCAN) (top right). Non-parametric permutation testing using Spearman rank correlations (two-tailed, no correction for multiple comparisons) revealed a significant positive relationship between tic reduction and connectivity with the CON ($r = 0.62$, $p = 0.0002$) and SCAN ($r = 0.47$; $p = 0.002$), indicating that greater connectivity between the DBS sites and these networks is associated with more substantial reductions in tic severity. **b** Compared to other large-scale resting-state, only CON and SCAN showed a significant positive association with tic reduction (permutation test for two-tailed Spearman rank correlations, corrected for multiple comparisons with $p_{FDR} < 0.05$; marked with asterisks). **c** A post-hoc comparison revealed strong and significant associations between tic reduction after DBS and connectivity with the

inferior ($r = 0.596$; $p = 0.0002$) and middle ($r = 0.513$; $p = 0.0002$) inter-effector regions, while connectivity to the superior inter-effector region showed a weak association ($r = 0.276$; $p = 0.052$) (permutation tests for two-tailed spearman rank correlations, not corrected for multiple comparisons). **d** Computing a data-driven, whole-brain voxel-wise model of optimal brain connectivity for tic reduction, termed R-map, clusters with a significant ($p_{FDR} < 0.05$) positive association between connectivity and tic reduction (red clusters) closely aligned with the CON. Within M1, the middle and inferior inter-effector regions of the SCAN were involved. Significant negative associations between connectivity and tic reduction are indicated in blue. **e** Subcortically, the insular clusters of the Tic-Reduction R-map further encompassed the posterior claustrum and small parts of the posterior putamen ($z = 1$). All scatter plots include linear trend lines with 95% confidence bounds. $a$ = anterior; $p$ = posterior; $l$ = left; $r$ = right.

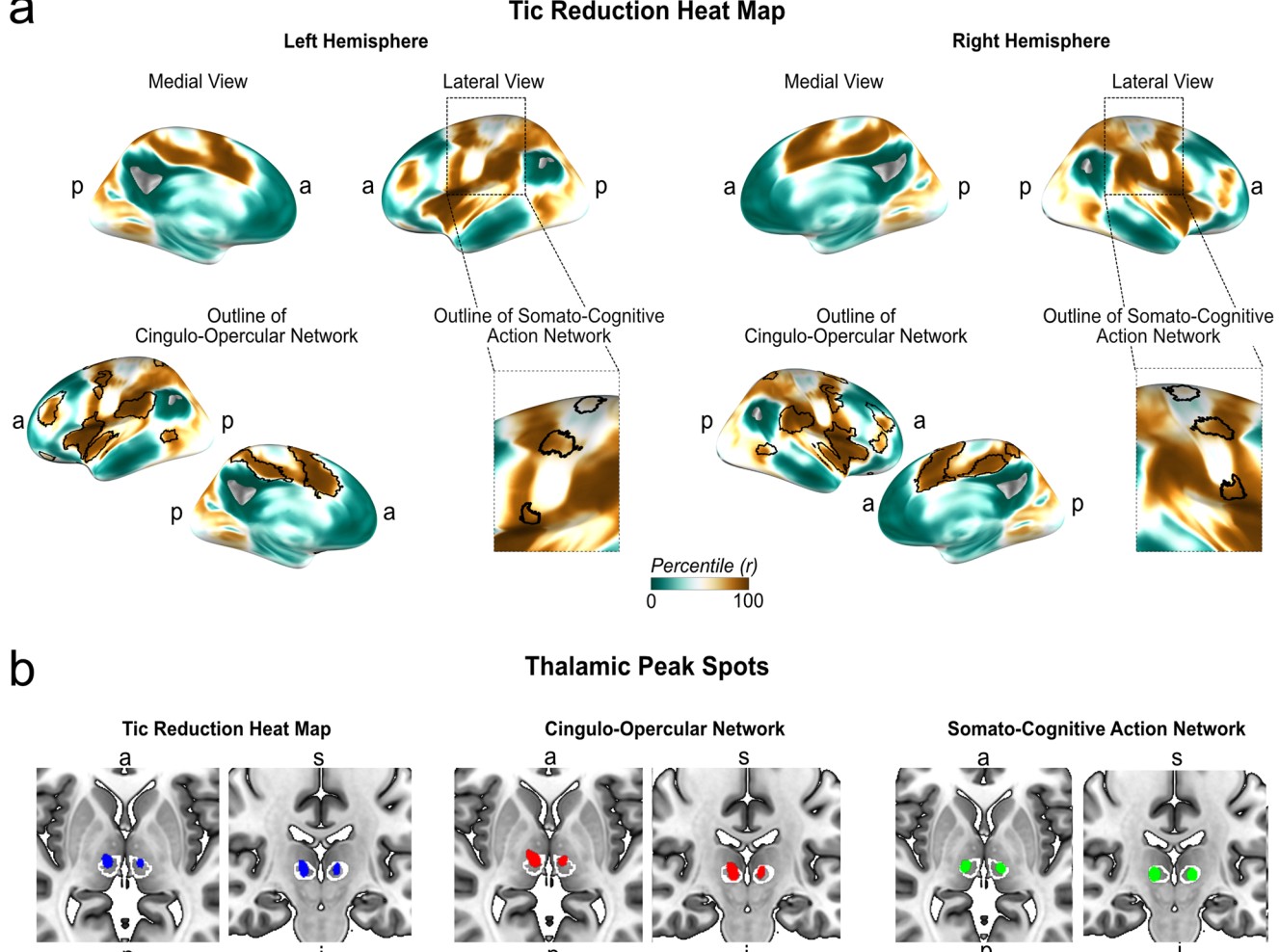

**Fig. 4 | Target heat map for tic reduction. a** We computed a target heat map containing the voxel-wise connectivity to the R-map to identify a network profile that matches the optimal connectivity profile derived from thalamic stimulation ($N = 37$). The highest matches were observed within the cingulo-opercular network (CON). Within the motor cortex, the somato-cognitive action network (SCAN) showed the highest similarity of connectivity with the tic-reduction R-map. **b** Within the thalamus, the heat map peaked in the superior and anterior part of the centromedian nucleus (outlined in white). Connectivity of the cingulo-opercular network and the somato-cognitive action network also showed the highest connectivity in the centromedian nucleus, highly overlapping with the peak connectivity of the target heat map (all three maps are thresholded at $r \geq 0.2$; slices taken at $z = 1$ and $y = 20$). $a$ = anterior; $p$ = posterior; $s$ = superior; $i$ = inferior.

outcomes after DBS. Both in the a priori-defined ROI analysis and the whole-brain data-driven approach, connectivity with the CON and SCAN emerged as strong indicators of treatment success. In hindsight, our previous analysis of a subset of 15 patients already showed a connectivity profile resembling the CON distribution[23]. The current study validates this finding in a larger multicenter dataset with

additional replications and establishes an additional role of the newly uncovered SCAN[13]. Within the SCAN, a post-hoc analysis showed that the inferior and middle inter-effector regions predominantly contributed to the overall positive association of tic reduction and functional connectivity of stimulation sites. While the exact roles of the different inter-effector regions are not fully understood[13], their

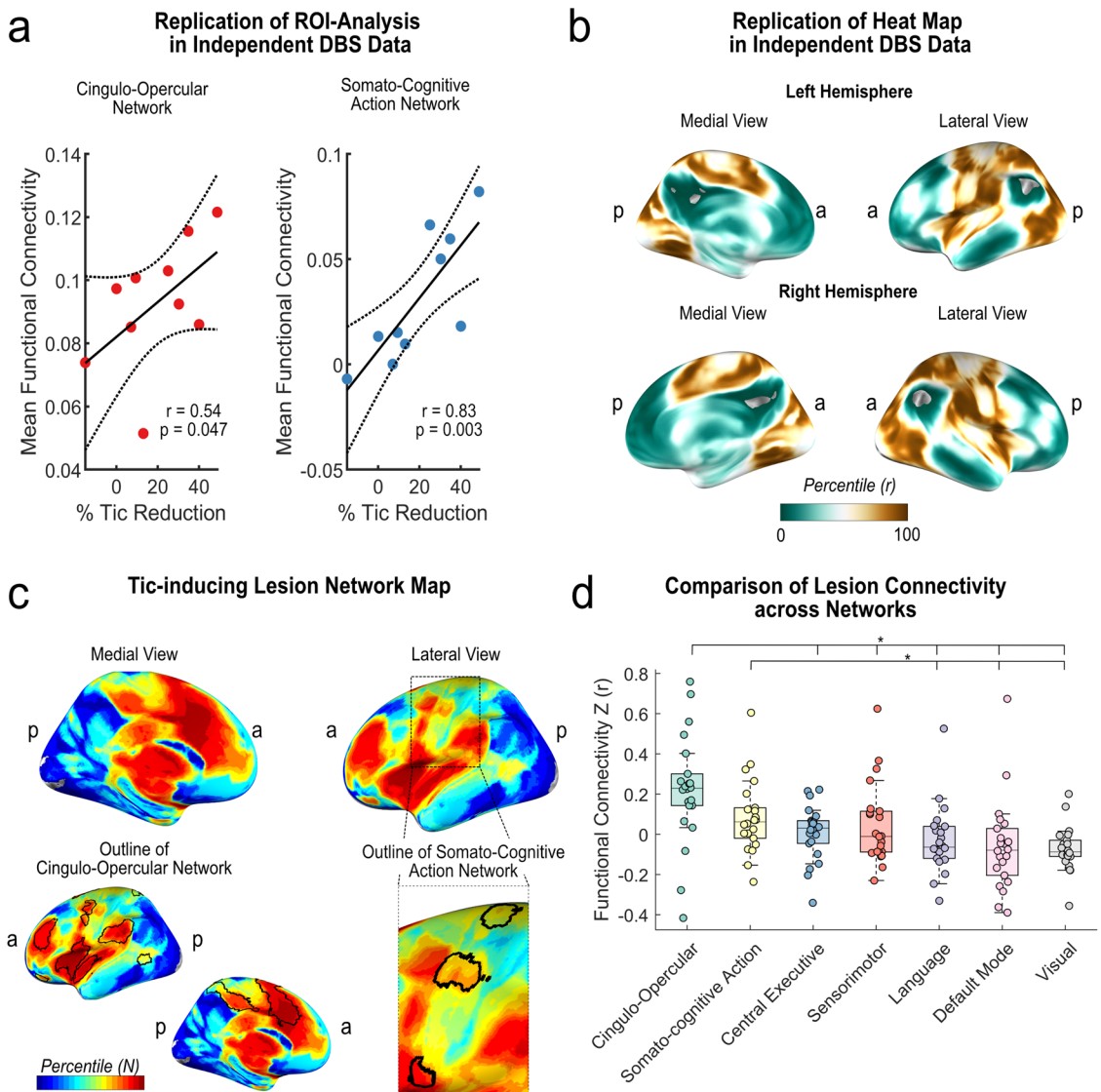

**Fig. 5 | Replication in independent data. a** In an independent dataset of pre-computed DBS sites ($N = 10$) obtained from the GTS-DBS-Registry, tic reduction was also significantly and positively correlated with functional connectivity (FC) to the cingulo-opercular (CON) ($r = 0.054$; $p = 0.047$) and somato-cognitive action network (SCAN) ($r = 0.83$; $p = 0.003$) (permutation-based two-tailed spearman rank correlation, not corrected for multiple comparisons). **b** Similarly, CON and SCAN became apparent in the heat map derived from this sample. **c** As a further replication, we investigated a previously published lesion network map. In the unthresholded lesion network map, derived from $N = 22$ lesions that induced tics, CON and SCAN were traceable as networks to which most lesions were connected. **d** The box plot displays the functional connectivity of each tic-inducing lesion to each large-scale resting-state network. Here, CON and SCAN showed the highest connectivity with respective lesions. Friedman's test with post-hoc pairwise comparisons revealed that CON had significantly higher connectivity than all other networks except SCAN (two-tailed tests, adjusted for multiple comparisons with pFDR < 0.05). Asterisks indicate significant differences in connectivity. All results were highly similar for both hemispheres; only left-hemispherical results are shown for clarity. All scatter plots include linear trend lines with 95% confidence bounds. Box-plot elements include the centre line representing the median, box limits showing the upper and lower quartiles, whiskers extending to 1.5 times the interquartile range, and individual points. a = anterior; p = posteriorFigure 6: Peak Clusters of Target Heat Map. The top five clusters of the target heat map for tic reduction are shown (yellow, as derived when thresholding voxels at $r > 0.6$). The clusters peaked in the bilateral insula/operculum (left image, $z = 3$), the right supplementary motor area (SMA) (middle image, $x = 6$), and the bilateral supramarginal gyrus (right image, $z = 27$). All clusters resided within the cingulo-opercular network, shown in purple. The bottom of the figure shows critical evidence supporting the relevance of these areas for tic disorders in general and as neuromodulation targets[52–58]. OLT = Open-label trial; RCT = randomized clinical trial; tDCS = transcranial direct current stimulation; fMRI = functional magnetic resonance imaging; rTMS = repetitive transcranial magnetic stimulation; a = anterior; p = posterior; l = left; r = right; sign. = significant.

involvement may reflect the somatotopic organization of the sensorimotor cortex. This organization would align with the distribution of tics in GTS, where most tics involve the head and face muscles, followed by the hand/shoulder muscles, with fewer tics involving the feet and body core[30]. Similarly, the premonitory urge to tic is most commonly experienced in the head/face region, followed by the upper extremity, and is least experienced in the lower extremities[31–38]. Therefore, although speculative at this moment, the differential engagement of the inferior, middle, and superior inter-effector regions observed in our tic reduction R-map might reflect the somatotopic distribution of tics and premonitory urges.

Exploration of the functional connectivity in normative resting-state data can potentially uncover polysynaptic networks that moderate tic reduction beyond the DBS target regions. Indeed, our analysis implicates widespread cortical connections involving motor/premotor, sensory and limbic networks that closely resemble the action-

related CON[15] and SCAN[13]. Comparing our results based on functional connectivity to the available evidence on structural connectivity of DBS for GTS leads to several observations that align with our findings. Johnson et al. revealed that structural connections derived from a normative connectome from the DBS site to the sensorimotor cortex, including areas of the CON and SCAN, were predictive of patient outcomes from the GTS-DBS-Registry[39]. Recently, Avecillas-Chasin et al. retrospectively investigated subcortical white matter pathways mediating tic reduction in a cohort of 21 patients who underwent either pallidal or thalamic DBS for GTS using a normative structural connectome based on 72 subjects[40]. The authors delineated two main networks, i.e., the limbic pallidothalamic network for the pallidal target, and a premotor thalamo-cortical projection for the thalamic target. For the thalamic target, tic reduction was most substantial when fibres to the SMA/DMPFC were targeted. This observation further matches an investigation from Andrade et al. using patient-specific diffusion magnetic resonance imaging in a small sample of patients with GTS ($N = 7$) undergoing thalamic DBS, showing that treatment outcomes related to an increased density of projections to the SMA/DMPFC[41]. For the pallidal target, Avecillas-Chasin et al. identified a prefrontal pathway to limbic areas *and* connections to the thalamic target, i.e., the CM/VOI. The authors concluded that this dual pathway model may constitute a limbic-motor interface network. Johnson et al. delineated a limbic/associative pallido-subthalamic pathway[42] as an additional circuit mechanism for the pallidal target. Our results expand the notion that prefrontal/limbic and motor networks are pivotal for DBS at the functional level. The prominent role of the CON and SCAN in our results further provides a link to understanding how motor and prefrontal/limbic networks interact in effective neuromodulation for tic disorders. Within the thalamus, the target heat map peaked in the anterior and dorsal part of the CM, suggesting that this region may represent the optimal stimulation target based on our analysis. However, this finding should be interpreted with caution due to the inherent limitations in spatial resolution associated with functional connectivity analyses. While the commonly used resolution of 2 x 2 x 2 mm in resting-state fMRI is adequate for seeding stimulation sites and investigating whole-brain functional networks, it may not be sufficient to precisely determine optimal target regions. An alternative approach would involve applying voxel-wise analyses of the electric field or the volume of activated tissue to generate probabilistic stimulation maps that predict therapeutic outcomes. Indeed, two previous studies employing such probabilistic stimulation maps found that targeting the anterior dorsal CM yielded optimal results, consistent with our findings[43,44]. However, an earlier analysis failed to identify a clear pattern using such probabilistic maps, potentially due to extended and heterogeneous follow-up periods of outcome assessments[45]. Therefore, further research with larger samples and well-controlled outcome assessments are required to accurately delineate the optimal thalamic target for DBS. A further important step could be to investigate whether similar functional networks, as identified in the current analysis are critical for DBS of the globus pallidus internus.

Beyond hypothesising about targeting functional connections to the CON and SCAN, we sought to determine whether our dataset could inform us about cortical areas potentially relevant for non-invasive neuromodulatory approaches. Based on the connectivity profile of successful tic reduction, we derived an inverse target heat map depicting cortical regions with similar resting-state connectivity. The peak voxels of this target heat map resided in the SMA, operculum/insula, and supramarginal gyrus. Strikingly, emerging evidence from non-invasive neuromodulation of exactly these three target regions already corroborates potential therapeutic effects in tic disorders. The SMA constitutes the most common target and has shown promising results in studies using transcranial magnetic stimulation (TMS)[46–49], although early randomized clinical trials did not show efficacy[50,51].

Transcranial direct current stimulation (tDCS) to the SMA reduced tic severity in a sham-controlled study[52]. Further, a randomized, sham-controlled trial of fMRI-guided neurofeedback intervention employing neural activity in the SMA effectively reduced tics in adolescents with GTS[53]. Fu et al. demonstrated that targeting the supramarginal and angular gyrus with TMS notably improved tic severity in another sham-controlled clinical trial[54]. The anterior dorsal insula was the third prominent region of our tic-reduction target heat map. fMRI studies have shown that the anterior insula is hyperconnected with the SMA in patients with tic disorders and that this connectivity correlates with urge severity[11]. Consequently, the insula, and specifically insula-SMA hyperconnectivity, has been repeatedly suggested as a potential target network underlying effective rTMS to the SMA[55,56]. Thus, our data-driven target heat map for tic reduction is consistent with the most promising known non-invasive targets for tic disorders and their presumptive underlying target network (Fig. 6). The exact three main clusters, i.e., insula, SMA, and supramarginal gyrus, were even suggested as most promising targets in a recent review article on non-invasive brain stimulation for tic disorders[56]. Our analysis provides unifying evidence characterising these target areas as critical parts of a shared network. Collectively, this network represents core areas of the CON network. The SCAN was also identifiable within the target heat map, although effect sizes were substantially higher within the CON. An early study showed no effect of TMS on the effector regions of M1 in tic disorders[57]. Whether neuromodulation to the newly uncovered SCAN may interfere with tic execution remains to be answered.

Our results and the existing literature on non-invasive brain modulation with its assumed underlying network mechanisms provide strong arguments that the CON-SCAN axis constitutes a potential target network for neuromodulation in tic disorders. Regarding DBS, prospective validation of these findings could be accomplished by individually refining stimulation parameters in a way that they show the highest connectivity with CON and SCAN. Regarding non-invasive brain stimulation, the target areas identified in our target heat map could be translated as multifocal targets for non-invasive neuromodulation protocols. Such an approach was recently successfully studied in a clinical trial for Parkinson's disease (DRKS00026640)[58].

Our observation that targeting CON and SCAN via thalamic DBS is associated with tic reduction, poses the question of the relevance of these networks in the pathophysiology of tic disorders. The observation that not only effective treatment in the form of DBS is connected to these networks but that tic-inducing lesions show similar connectivity suggests that these networks are indeed critically involved in the emergence of tics. The CON, also called action-mode network, combines information from the environment to initiate actions that can range from mental processes to motor actions[14]. The premonitory urge to tic may resemble a phenomenological representation of this function – an internal command that facilitates behaviour that can range from simple motor actions (e.g., simple motor tics such as eye blinking) to sequences of behaviours (e.g., complex tics like touching), but may also include mental phenomena (e.g., obsessive-compulsive behaviour like counting). Indeed, premonitory urge intensity has been linked to neural activity in critical CON areas, i.e., insula, SMA/dACC and parietal cortex[3,9,11,21,22]. In a recent preprint article, D'Andrea et al. were able to form functional subdivisions of the CON that could be attributed to either sensory feedback, decision or action[17]. By analysing the temporal order of activation, D'Andrea et al. further proposes a hierarchical temporal model for action control, where the decision-CON receives information about aversive stimuli, followed by activity in the action-CON that initiates a response and subsequently in the SCAN as an entry-point to the effector regions[17]. Our analysis suggests that the tic reduction network predominantly corresponds to the decision and action subdivisions of the CON. The SCAN may then constitute an integrative role in tic disorders that facilitates the transition from urge-related neural activity in the CON to an action, i.e., the

# Peak Clusters of Target Heat Map

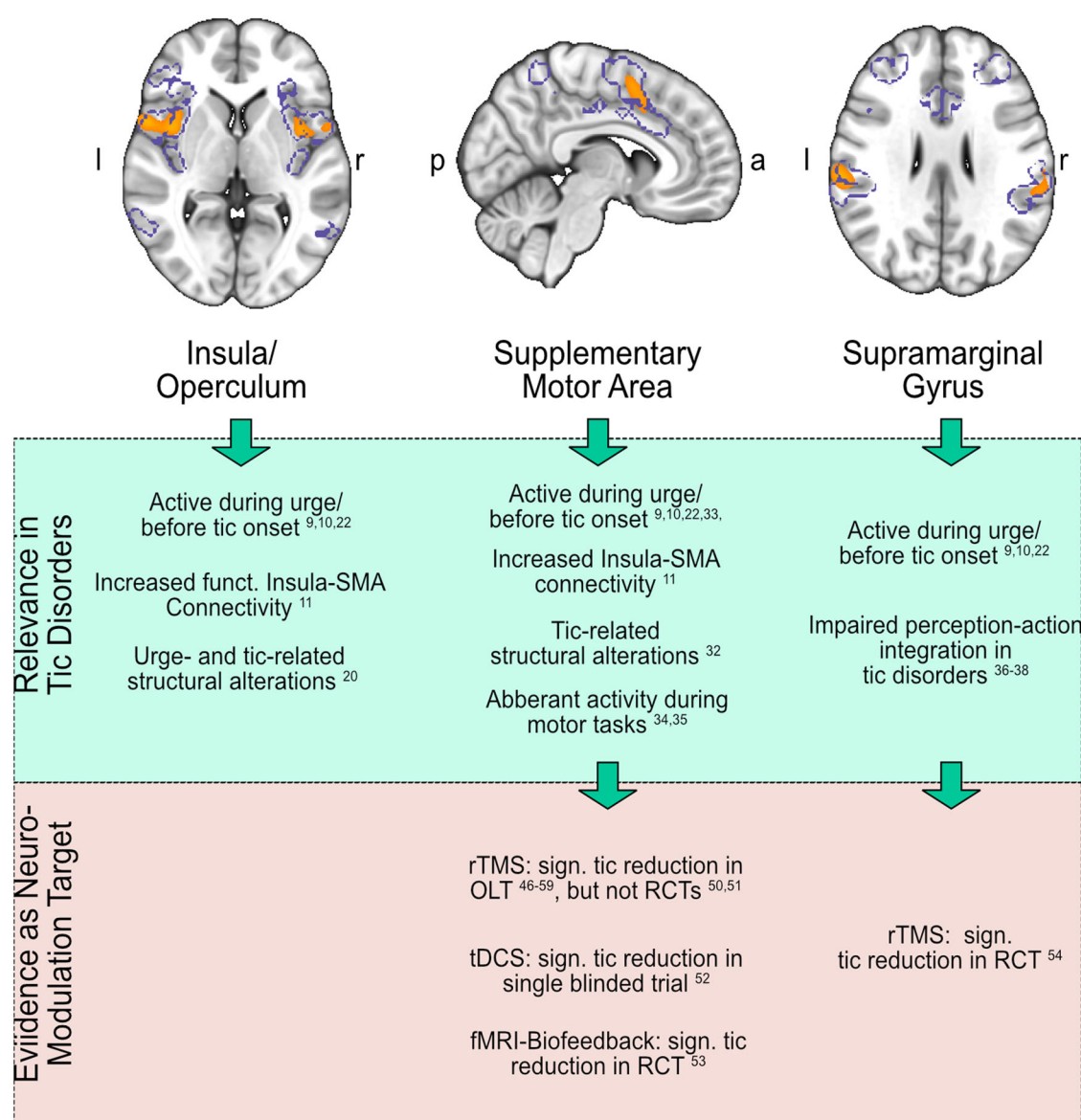

**Fig. 6 | Peak clusters of target heat map.** The top five clusters of the target heat map for tic reduction are shown (yellow, as derived when thresholding voxels at $r > 0.6$). The clusters peaked in the bilateral insula/operculum (left image, $z = 3$), the right supplementary motor area (SMA) (middle image, $x = 6$), and the bilateral supramarginal gyrus (right image, $z = 27$). All clusters resided within the cingulo-opercular network, shown in purple. The bottom of the figure shows critical evidence supporting the relevance of these areas for tic disorders in general and as neuromodulation targets[52–58]. OLT = Open-label trial; RCT = randomized clinical trial; tDCS = transcranial direct current stimulation; fMRI = functional magnetic resonance imaging; rTMS = repetitive transcranial magnetic stimulation; $a$ = anterior; $p$ = posterior; $l$ = left; $r$ = right; sign. = significant.

tic. Future studies may follow up on this hypothesis using patient-specific neural activity data during urges and tics.

Several limitations have to be considered when interpreting our results. First, even though the current analysis constitutes one of the largest cohorts of DBS for tic disorders, the global scarcity of this patient collective leads to a limited sample size. However, we confirmed the link between CON/SCAN connectivity and tic reduction in two independent DBS samples and found consistent heat maps across DBS and lesion cohorts. Nonetheless, further validation of these findings is imperative, e.g., by probing the individualized targeting of these networks. Further, the current study used a normative functional connectome that was derived from 1000 healthy participants, thereby neglecting potential disease-specific connectome disruptions or individual differences in connectivity. This approach may serve as a broader characterisation of the underlying stimulated or lesioned networks in an averaged, healthy brain, where the normative connectome can be understood as an atlas of average brain connectivity. This normative method has previously effectively predicted DBS

treatment responses across various conditions[26,27,59]. Beyond DBS, neural networks of numerous neuropsychiatric symptoms caused by heterogeneous focal brain lesions could be explained using the same fMRI data[28,60–63]. Our investigation of lesions causing tic symptomatology matched the connectivity profile derived from DBS, indicating promising generalisability of the tic reduction map across brain interventions despite the use of normative connectivity data. Nonetheless, patient- or disease-specific information about brain connectivity may allow to explain more variance in DBS outcomes. This is particularly of interest since GTS constitutes a neurodevelopmental disorder. Thus, it could be possible that in people with GTS, the investigated networks develop differently across the lifespan. Indeed, an earlier study suggests that the CON shows altered maturation in adolescents with GTS compared to age-matched healthy controls[18]. A similar conclusion, i.e., altered maturation of functional brain networks in GTS, was drawn from a more recent study. Here, a machine learning-derived classification of GTS using functional connectivity networks successfully distinguished patients with GTS from healthy controls – however, the distinguishing features differed between children and adults, pointing towards different neurodevelopmental trajectories of functional brain networks in children and adults with GTS[64]. That being said, the principal functional brain networks were well traceable in the patient cohorts in both studies. Thus, although quantitative differences in these networks may be assumed, it appears unlikely that the general architecture of functional brain networks is drastically different in patients with GTS. Along these lines, a study of patients undergoing DBS for Parkinson's disease showed that the usage of patient-specific connectivity was not significantly superior to normative connectivity data and that both approaches revealed similar optimal brain networks to be targeted[65]. Nonetheless, given the potential disease-specific brain alterations, we assume that at least some more variance of outcomes may be explainable by using disease-specific or even patient-specific connectivity data. However, obtaining high-quality resting-state fMRI data from unsedated, severely affected patients with tic disorders that may qualify for DBS is hardly feasible, meaning that this approach is currently the only way to investigate functional connectivity estimates in patients with tic disorders who received DBS in a larger sample. One potential solution for this limitation could be to investigate a disease-specific functional group connectome derived from patients with less severe tics, where rs-fMRI of sufficient quality can be assessed. With such a group connectome, it could be tested whether the predictability of outcomes could be improved compared to the currently used normative group connectome. The current analysis aimed to determine whether the connectivity of stimulation sites to specific functional networks could explain the variability in treatment outcomes. Ideally, this approach would be enhanced by incorporating additional clinical or biological predictors that could provide insights into the likelihood of responding to DBS surgery. Employing such predictors could also help refine the targeting of networks on a personalised basis. However, here and also, to the best of our knowledge, in the available literature to date, no preoperative predictors could be identified for GTS. Regarding the lesion network map, we had no access to control lesions, i.e., equivalent lesions inducing symptoms other than tics. Thus we cannot answer whether other symptoms may as well result from lesions to these networks. Further, the treatment effects of DBS were assessed by calculating the change in tic severity from a single pre- to postoperative assessment of the YGTSS. Since the severity of tics naturally waxes and wanes over time, multiple assessments would have been advantageous to account for baseline variance in tic severity. However, as stimulation settings were subject to changes across clinical visits, only one assessment per stimulation setting was available across the cohort. Although this may introduce some variability on an individual level, the relatively large sample size in our study helps to minimize potential distortions in the overall findings.

In conclusion, our findings consistently demonstrate that thalamic DBS yields the highest efficacy when targeting sites connected to the CON and SCAN. Similar network patterns were observed in lesions that induce tics, underscoring the pivotal role of these networks in the pathophysiology of tic disorders and their therapeutic modulation. Furthermore, we have identified critical regions within the CON that constitute a cortical target network, which could serve as a roadmap for guiding both invasive and non-invasive neuromodulatory interventions for treating tic disorders.

## Methods
### Patient cohort and imaging
The study was carried out in accordance with the Declaration of Helsinki and approved by the local ethics committee at the University Hospital Cologne under approval number 23-1409-retro. We sourced data from adult patients with severe GTS undergoing DBS ($N = 37$) in three European centres: Cologne, Germany ($n = 21$); Milan, Italy ($n = 8$); and Hannover, Germany ($n = 8$). In addition, the International Tourette Deep Brain Stimulation Database and Registry (henceforth, GTS-DBS-Registry)[66] provided data for further replication ($N = 10$). In the Cologne cohort, 12 of 21 patients participated in two prospective clinical trials[43,67]. From the first trial[67], four patients lacked postoperative imaging adequately allowing electrode reconstruction in standard space, while from the second, one patient was excluded due to a suspected combination of GTS and functional tic-like behaviour (FTLB) as previously described[43]. The Hannover cohort, derived from a further recent clinical trial, excluded two patients with GTS also suffering from comorbid FTLB as outlined in the original publication[68]. All patients gave informed consent to take part in the respective studies. Milan's cohort was based on retrospective data of nine patients who received treatment following individual clinical decision-making. In this dataset, one patient had to be excluded due to insufficient postoperative imaging data. For external replication, we referenced previously published data from patients of the GTS-DBS-Registry who underwent thalamic DBS[39]. From this set, we excluded data whose source data was already used in this study or which was assessed outside the six to twelve months post-intervention, leaving ten patients.

All patients received DBS surgery that targeted subnuclei of the thalamus bilaterally using two quadripolar electrodes (Medtronic 3387 or 3389, Medtronic, Minneapolis, MN). In the Cologne cohort, four electrodes were implanted in the ventral anterior (VA) and ventrolateral (VL) nucleus of the thalamus with the distal contacts residing in the field of Forel/subthalamic area. All other electrodes were implanted in the CM and the nucleus ventro-oralis internus (VOI)[67]. Notably, the VOI is part of the VL, designated by different terminologies[69]. The variability in the location of stimulation sites, e.g., due to precise presurgical targets and trajectories, electrode placement, and the spread of the electric field was leveraged to compute brain networks explaining optimal treatment outcomes and thus must be seen as an advantage for this study. All subjects received preoperative T1 volumetric magnetic resonance imaging (MRI) scans. Electrode placements were confirmed using postoperative MRI or computer tomography (CT) scans (see Supplementary Table 2 for specification of pre- and postoperative imaging parameters). We used the percentage change in the Yale Global Tic Severity Scale (YGTSS)[70] global score (comprising the total tic score and impairment score) as the primary outcome parameter, which reflects the severity of tics over the past week. In three patients from the Milan cohort, the impairment score was not available, and thus only the total tic score was used. Of note, the global and total tic scores show a highly similar correlation with the clinically evaluated disease severity and thus serve as comparable outcome parameters[71]. As follow up, the earliest single assessment in a period between six to twelve months post-intervention was chosen. To investigate whether clinical and demographic factors

could account for variations in treatment outcomes, we conducted a Spearman correlation analysis between patient age and the percentage change in YGTSS scores. Furthermore, we employed Wilcoxon rank-sum tests to compare treatment outcomes between female and male patients, as well as between patients with and without comorbid obsessive-compulsive behaviour. The presence of obsessive-compulsive behaviour was assessed either through clinical transcripts or a minimum score of 7 on the Yale-Brown Obsessive Compulsive Scale[72].

### Computation of electrode reconstructions, stimulation sites and functional connectivity

Electrode reconstructions and stimulation sites were processed using Lead-DBS software[73] (Version 2.6, www.lead-dbs.org). Postoperative scans were aligned with preoperative MRI images using SPM 12 for MRI and Advanced Normalization Tools[74] (ANTs, http://stnava.github.io/ANTs/) for CT scans. A subcortical refinement step in Lead-DBS addressed potential brain shifts during surgery. Images were then normalized to the Montreal Neurological Institute (MNI) 2009b non-linear asymmetric template space, utilizing the symmetric normalization method in ANTs and the subcortical refinement tool from Lead-DBS. Electrode positions were determined by the PaCER algorithm[75] or the TRAC/CORE method within Lead-DBS. Electric fields were estimated based on individual stimulation settings using FastField[76] within Lead-DBS. Stimulation sites, i.e., the binarized volume of activated tissue (VAT), were derived by thresholding the electric field at 0.2 V/mm, according to work by Åström et al. [77]. Importantly, in the GTS-DBS-Registry dataset the stimulation sites were independently modelled in each patient[39] and remained unmodified for the present analysis to test the generalisability of the results.

Functional connectivity maps and subsequent statistical analysis were performed in Lead-DBS v2.6[73] and using customised scripts within MATLAB R2022b (The MathWorks Inc., Natick, MA, USA). Patient-specific functional connectivity maps were constructed using the Lead Mapper function in Lead-DBS[73], leveraging a public resting-state functional MRI dataset acquired in 1000 healthy participants from the Brain Genomics Superstruct Project (https://dataverse.harvard.edu/dataverse/GSP)[78]. Functional images were acquired using a 3 Tesla Siemens scanner with a resolution of $2 \times 2 \times 2$ mm. Preprocessing of the blood oxygen level-dependent time courses included regression of global signal, white matter, and cerebrospinal fluid signals, as well as the six motion parameters. Smoothing was performed using a 6 mm full-width at half-maximum (FWHM) Gaussian kernel, as previously described[78]. For each bilateral pair of stimulation sites, the averaged functional connectivity across the 1,000 subjects to the rest of the brain was computed by sampling time series from voxels within the binary stimulation sites in each subject of the normative connectome and correlating these time series with those of every other voxel. This procedure resulted in a stimulation-dependent functional connectivity map for each patient, containing Fisher-z-transformed connectivity strengths that were used for subsequent group analysis (Fig. 1a).

Notably, other groups[59] utilized the unthresholded electric field as a weighted seed to compute stimulation-specific connectivity maps for subsequent group analysis instead of using the binarized stimulation site. To test whether this could have influenced our results, we replicated the same analysis employing the unthresholded electric field instead of binarized stimulation sites. As illustrated in Supplementary Fig. 1, this produced highly comparable results.

### Computing functional connectivity associated with tic reduction

**Average whole-brain connectivity and region-of-interest analysis.** An overview of the group analysis is given in Fig. 1b. In the first analysis, we computed the voxel-wise averaged connectivity across all bilateral pairs of stimulation sites, termed the average map. Performing a region of interest (ROI) analysis, we tested whether tic reduction after DBS was associated with functional connectivity to predefined networks derived from resting-state fMRI. These networks comprised the CON and SCAN, as recently introduced by Gordon et al. [13]. The authors of the original publication graciously shared the respective ROI files. For each subject, we calculated the connectivity of each bilateral pair of stimulation sites with the respective network ROI. The averaged connectivity was then correlated with the clinical outcomes (percent improvement in YGTSS scores) using Spearman correlations. Statistical significance was determined using non-parametric permutation testing with 5000 iterations and a p-threshold of 0.05[26]. This method does not rely on assumptions regarding distributions (such as Student's t for r values), which typically do not hold true in situations with small sample sizes. In testing the specificity of results, we additionally correlated tic reductions with connectivity to other major functional resting-state networks derived from the Consensual Atlas of resting-state Networks (CAREN)[79], i.e., the sensorimotor, central executive (often referred to as frontoparietal network), visual, language and default mode network. A Benjamini & Hochberg adjustment for controlling the false discovery rate (FDR)[80] was applied ($p_{FDR} = 0.05$) to correct for multiple comparisons. Finally, we conducted a post-hoc analysis to investigate how the three different inter-effector regions (inferior, middle, and superior) contributed to the treatment outcomes. We calculated the connectivity of DBS sites to each inter-effector region and separately correlated these estimates with the percentage reduction in tic severity.

**Whole-brain R-map.** In addition, we aimed to calculate a data-driven functional connectivity map associated with tic reduction after DBS. To this end, we calculated a whole-brain voxel-wise R-map as described in previous studies[23,26,81]. For each grey matter voxel, the functional connectivity estimates were correlated with tic reduction scores across subjects using Spearman correlations. This map was thresholded using an adjusted $p_{FDR} = 0.05$ and a cluster-extent threshold of $k = 10$ voxels. In this map, high positive values indicate a strong association between connectivity with the target region and higher tic reduction, while high negative values indicate an association with less tic reduction. We quantified the overlap of the R-map indicative of tic reduction with our ROI networks CON and SCAN, as well as with the previously mentioned functional brain networks derived from the CAREN atlas[79] (sensorimotor, central executive, visual, language and default mode network). In addition, we calculated the balanced accuracy as the mean of sensitivity and specificity, offering a balanced evaluation of true positive and true negative rates. Sensitivity (True Positive Rate) was calculated as the ratio of overlapping voxels (voxels in the positive R-map that also belong to the network) to the total number of voxels in the network (overlapping + non-overlapping voxels in the network). Specificity (True Negative Rate) was calculated as the ratio of non-overlapping voxels outside the network (voxels not in the positive R-map and not in the network) to the total number of non-network voxels (non-overlapping + overlapping voxels outside the network). Balanced accuracy was then determined by averaging the sensitivity and specificity, providing a robust metric that accounts for both the true positive and true negative rates. Generally, balanced accuracy ranges from 0 to 1, with 0.5 indicating a classification at chance level and 1 representing a perfect fit.

**Target heat map for tic reduction.** Finally, we aimed to create a target heat map to potentially guide novel optimal stimulation targets for tic disorders, inspired by prior work in addiction[28], OCD[26], and Parkinson's disease[27]. To this end, the whole-brain connectivity map for tic reduction was inversed by calculating the weighted functional connectivity of the previously computed, FDR-corrected whole-brain map of functional connectivity associated with tic reduction with each

other grey matter voxel. In this target heat map, we display percentiles representing each voxel's likelihood of matching the cortical connectivity profile of the primary tic reduction network derived from subcortical thalamic stimulation, thereby potentially identifying therapeutic targets for neuromodulation. When using the R-map as a seed for this analysis, we masked the subcortex to avoid a circular analysis that would be overly similar to the analysis of thalamic stimulation sites. Notably, masking only the thalamus led to highly similar results (see Supplementary Fig. 2). Peak clusters of the target heat map were obtained by thresholding all voxels below a strong association ($r < 0.6$) and a minimum cluster size of 10 voxels.

**Replication in independent datasets.** To test the robustness of the results, we employed the GTS-DBS-Registry sample. In this sample, electrodes and stimulation sites were independently modelled as described in the respective publication[39] to avoid any bias that could result from recomputing the dataset. First, we conducted a similar ROI analysis using the CON and SCAN networks. Second, we computed a similar target heat map. Given the small sample size in the replication cohort ($n = 10$), the R-map computed in this sample was not thresholded to avoid Type II errors. The unthresholded map was then used to create the target heat map. Because this R-map comprised subcortical voxels including the thalamus, we restricted this analysis to cortical voxels to avoid circularity of analysis.

To investigate whether the relevance of CON and SCAN holds true in datasets beyond DBS, we additionally investigated a previously published lesion network map[24]. Regarding lesion network map computation, we refer to the original publication. To briefly summarize, a set of $N = 22$ brain lesions that led to the occurrence of tics were used to compute a brain-wide functional connectivity map. Lesions were extracted from scientific publications reporting new-onset tics attributed to central nervous system lesions. Lesion locations were manually traced from the corresponding publication figures and then used as seeds for the Lead Connectome Mapper toolbox within Lead-DBS[73]. Functional connectivity of the seeds was calculated using the same connectomic dataset of 1000 healthy participants used in the present analysis[78]. From the resulting connectivity maps, voxel-wise one-sample T-Tests of all connectivity values were performed to obtain an average connectivity map of T-scores which was then thresholded based on previous publications (in detail discussed here[82]) to obtain a binarized map for each case where each voxel is defined as connected or unconnected with the respective lesion. In the original publication by Ganos et al., a group-level N-map was then created that only included voxels connected to 86% of cases to obtain a maximally common connectivity profile. Here, we explored the unthresholded N-map to investigate whether other regions may be connected to smaller numbers of tic-inducing lesions. For clarity, the N-map was transformed into percentage values, i.e., the percentage of lesions connected with a respective voxel from all tic-inducing lesions. While this approach may lack the specificity of inferred regions aimed for in the original publication[24], it increases the sensitivity to detect regions potentially involved in the de novo occurrence of tic disorders. Last, we tested whether lesions show higher connectivity with CON and SCAN compared to other major functional resting-state networks[79], similar to the analysis of the DBS cohorts. To this end, we performed a Friedman test between all connectivity values across networks for each lesion and a post-hoc comparison using the multcompare function in Matlab R2022b between CON and SCAN and further major functional resting-state networks (i.e., sensorimotor, central executive, visual, language and default mode network according to CAREN[79] atlas) using an $p_{FDR} = 0.05$ to control for multiple comparisons.

**Reporting summary**
Further information on research design is available in the Nature Portfolio Reporting Summary linked to this article.

## Data availability
Source data for all figures, except those visualizing imaging findings, are provided with this paper in a Source Data file. The Source Data file includes all relevant raw data from scatter, bar, and box plots. The main group analysis imaging results are publicly available in the Open Science Framework (OSF) repository (https://doi.org/10.17605/OSF.IO/UPJVB). Individual patient imaging data cannot be openly shared due to privacy and data-sharing regulations. However, data can be requested under specific data use agreements from the primary investigators at each centre. For such requests, the corresponding author (JCB) can be contacted to initiate the process of obtaining access in accordance with institutional and regulatory guidelines. The functional connectivity data used in this work is publicly available under https://dataverse.harvard.edu/dataset.xhtml?persistentId=doi:10.7910/DVN/25833. Source data are provided in this paper.

## Code availability
All code used to preprocess the data including the estimation of individual connectivity maps is openly available within the Lead DBS software (https://github.com/leaddbs/leaddbs). Project-specific code to reproduce results and figures of this manuscript is openly available on OSF (https://doi.org/10.17605/OSF.IO/UPJVB).

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

## Acknowledgements

This study was funded by the German Research Foundation (CRC-1451, Project 431549029-C07, to J.C.B. and V.V.V.) and the Else Kröner-Fresenius-Stiftung (grant number 2022_EKES.23 to J.C.B.). A.H. was supported by the German Research Foundation (Deutsche Forschungsge

meinschaft, 424778381 – TRR 295), Deutsches Zentrum für Luft- und Raumfahrt (DynaSti grant within the EU Joint Programme Neurodegenerative Disease Research, JPND), the National Institutes of Health (R01 13478451, 1R01NS127892-01, 2R01 MH113929 & UM1NS132358), and the New Venture Fund (FFOR Seed Grant). JNPS was funded by the Cologne Clinician Scientist Program (CCSP) / Faculty of Medicine / University of Cologne, funded by the German Research Foundation (DFG, FI 773/15-1).

## Author contributions

J.C.B. and M.B. designed the study, analysed the data, and wrote the manuscript. T.D., T.S., J.N.P.S., G.B., C.v.L. and L.M. analysed the data and critically revised the manuscript. J.K., N.S., K.M.V., C.G., B.A., P.H., D.S., T.G., K.A.J., C.R.B., M.S.O., P.A., J.K., K.D., G.R.F. and V.V.V. collected the data and/or critically revised the manuscript. A.H. and M.D.F. provided the normative functional data and data analysis pipeline and further critically revised the manuscript.

## Funding

## Competing interests

C.G. was supported by the VolkswagenStiftung (Freigeist) and received lecture honoraria from the Movement Disorder Society. K.M.V. has received financial or material research support from DFG: GZ MU 1527/3–1 GZ MU 1527/3–2, and Almirall Hermal GmbH. She has received consultants and other honoraria from Canymed, Emalex, Eurox Group, Sanity Group, Stadapharm GmbH, Swiss alpinapharm, Synendos Therapeutics AG, Tetrapharm, and Triaspharm. She is an advisory/scientific board member for Branchenverband Cannabiswirtschaft e.V. (BvCW), Sanity Group, Synendos Therapeutics AG, Syqe Medical Ltd., and Therapix Biosciences Ltd. She has received speaker's fees from Almirall, Bundesverband der pharmazeutischen Cannabinoidunternehmen (BPC), Cogitando GmbH, Emalex, Grow, Medizinischer Dienst Westfalen Lippe, Noema, streamedup! GmbH, and Vidal. She has received royalties from Elsevier, Medizinisch Wissenschaftliche Verlagsgesellschaft Berlin, and Kohlhammer. She is an associate editor for "Cannabis and Cannabinoid Research" and an Editorial Board Member of "Medical Cannabis and Cannabinoids" und "MDPI-Reports" and a Scientific board member for "Zeitschrift für Allgemeinmedizin". A.H. reports lecture fees for Boston Scientific and is a consultant for FxNeuromodulation and Abbott. V.V.V. reports lecture fees for Medtronic and Boston Scientific. J.K.K. is a consultant to Boston Scientific, Medtronic, aleva and Inomed. N.S. has received financial and material research support from the research grant from the Medical University of Warsaw, the Polish Ministry of Health, the Polish Neurological Society, the Foundation for Polish Research, the European Stroke Organization, the American Academy of Neurology,

American Brain Foundation and Tourette Association of America. She has received honoraria from Biogen and 90 Consulting. G.R.F. serves as an editorial board member of Cortex, Neurological Research and Practice, NeuroImage: Clinical, Zeitschrift für Neuropsychologie, and Info Neurologie & Psychiatrie; receives royalties from the publication of the books Funktionelle MRT in Psychiatrie und Neurologie, Neurologische Differentialdiagnose, SOP Neurologie, and Therapiehandbuch Neurologie; receives royalties from the publication of the neuropsychological tests KAS and Köpps; received honoraria for speaking engagements from Deutsche Gesellschaft für Neurologie (DGN) and Forum für medizinische Fortbildung FomF GmbH. All other authors report no competing interests.

## Additional information

[1]Department of Psychiatry and Psychotherapy, Medical Center – University of Freiburg, Faculty of Medicine, University of Freiburg, Freiburg, Germany. [2]Department of Neurology, Faculty of Medicine and University Hospital Cologne, University of Cologne, Cologne, Germany. [3]Department of Neurosurgery, Hannover Medical School, Hannover, Germany. [4]Department of Neurology, University of Calgary, Calgary, Alberta, Canada. [5]Department of Psychiatry, Socialpsychiatry and Psychotherapy, Hannover Medical School, Hannover, Germany. [6]Department of Bioethics, Medical University of Warsaw, Warsaw, Poland. [7]Movement Disorder Clinic, Edmond J. Safra Program in Parkinson's Disease, Division of Neurology University of Toronto, Toronto Western Hospital, Toronto, Canada. [8]Department of Neurology, Charité-Universitätsmedizin Berlin, Corporate Member of Freie Universität Berlin and Humboldt-Universität zu Berlin, Berlin, Germany. [9]Department of Stereotactic and Functional Neurosurgery, Faculty of Medicine and University Hospital Cologne, University of Cologne, Cologne, Germany. [10]Neurosurgical Department, IRCCS Istituto Ortopedico Galeazzi, Milan, Lombardia, Italy. [11]Norman Fixel Institute for Neurological Diseases, University of Florida, Gainesville, FL, USA; Department of Neurology, University of Florida, Gainesville, FL, USA. [12]Department of Neurosurgery, University of Florida, Gainesville, FL, USA; J Crayton Pruitt Family Department of Biomedical Engineering, University of Florida, Gainesville, FL, USA. [13]German Center for Mental Health (DZPG), Partner Site Berlin, Berlin, Germany. [14]Cognitive Neuroscience, Institute of Neuroscience and Medicine (INM-3), Research Center Jülich, Jülich, Germany. [15]Center for Brain Circuit Therapeutics, Department of Neurology, Brigham & Women's Hospital, Harvard Medical School, Boston, MA, USA. [16]Department of Neurology, Charité – Universitätsmedizin Berlin, Berlin, Germany. [17]Einstein Center for Neurosciences Berlin, Charité – Universitätsmedizin Berlin, Berlin, Germany. [18]Department of Neurosurgery, Massachusetts General Hospital, Harvard Medical School, Boston, MA, USA. [19]Alexianer Hospital Cologne, Alexianer Köln GmbH, Cologne, Germany. [20]Department of Psychiatry, Faculty of Medicine and University Hospital Cologne, University of Cologne, Cologne, Germany. ✉e-mail: juan.carlos.baldermann@uniklinik-freiburg.de

