## [Transparent Peer Review file · Nature Communications]

A Critical Role of Action-Related Functional Networks in Gilles de la Tourette Syndrome

Corresponding Author: Dr Juan Carlos Baldermann

Version 0:

Reviewer comments:

Reviewer #1

(Remarks to the Author)

In this manuscript, "A critical role for Action-Related Functional Networks in Gilles de la Tourette Syndrome, the authors test the hypothesis that functional connectivity to action-related networks, investigated in a multicenter cohort study of patients undergoing thalamic DBS or having tic-inducing spontaneous lesions, would be associated with tic reduction.

The principle findings in the study are:

- 1) Greater tic reduction in patients receiving thalamic DBS is linked to higher functional connectivity of the DBS sites to action related networks (as measured by using the patient sites to quantify connectivity in a openly available dataset of 1000 healthy adults.
- 2) The findings were replicable in a separate, smaller dataset of DBS receiving patients, and
- 3) Tic-inducing lesions revealed a comparable network connectivity profile, implicating the same action related networks.

The authors have produced a well-written and (mostly) well-figured manuscript that addresses a deeply interesting hypothesis and takes full advantage of collaboratively, multicenter collected patient data as well as a fairly large public MRI dataset collected from healthy individuals.

In addition, the authors have capitalized promptly on very recent advances from precision functional mapping using resting state functional connectivity MRI that have reshaped in a quite remarkable way how we think about the representation of the somatic motor system in primary motor cortex. The so-called Somatic Cognitive Action Network constitutes a reframing of how primary motor cortex is organized.

An important takeaway from this work is that the reframing of the classic motor homunculus has implications that go beyond updating the schematic renderings of the motor homunculus dating back over a century --the organizational and functional features of the SCAN very likely have clinical significance. That insight would be impossible to attain without the emergence of deeper understanding of the organization of the brain's functional network architecture through cutting edge advanced neuroimaging and network informed analyses.

The methodology is clearly presented and sound, and is sufficiently detailed for others to reproduce the approach.

This paper, therefore, has broad implications beyond the relatively narrow scope of a pediatric onset movement disorder (though the implications for Tourette syndrome are certainly substantial). The work is exciting, thought provoking, and important.

Minor criticisms:

- 1) The paper, while crisply written, would benefit from figures and figure labeling more mindful of the non-expert reader. For example, a figure showing brain surface renderings of each of the functional networks examined would be very helpful. Such a figure would also help make the SCAN and CON overlays more understandable to the uninitiated. Figure 6 provides A, P, R, L labels (not mentioned in the legend), but the other figures do not provide such orienting information. Overall, the recommendation is to reexamine and edit the figures taking into account the broader readership of this journal.
- 2) The limitations section does include some language about the benefit of precision functional imaging approaches and

how it is untenable at this time to endeavor to highly sample patients with such severe ticcing. That said, the manuscript would benefit from a deeper dive into the topic. For example, some proportion of the variance assuredly comes from the fact that the presumptive functional connectivity of individual patient's electrode sites is being mapped to a population sample that is non highly sampled and placed in standard space. Again, this comment is not intended to be a criticism. Rather, the reader would benefit from understanding the potential consequences.

3) Along these lines, while precision mapping may not be tenable in patients with severe ticcing, it likely is tenable in patients with mild to moderate tics. The question of whether network architecture is disrupted developmentally in the context of Tourette syndrome, signal may be evident in such precision mapped individuals. Not a request for additional investigation.

4) The term "hub" is used multiple times in the paper. It is not clear that the term is used accurately with respect to its meaning in networks. Rather, it seems that "hub" is being used to mean "target".

(Remarks on code availability)

To my understanding, there is sufficient code information for replication.

Reviewer #2

(Remarks to the Author)

In this article, Baldermann and colleagues have described two cohorts of patients with Tourette syndrome who underwent DBS to reduce tic severity, and a cohort of patients with lesion-induced tics. The research team assessed resting state functional connectivity at DBS sites that were more vs. less effective at tic reduction and demonstrated that stimulation sites that were effective at reducing tics had significant functional connectivity with the CON and SCAN networks. They replicated these findings in previously-published cohorts of patients treated with DBS for tics and in patients whose brain lesions induced tics, demonstrating similar functional network correlations. These findings are of considerable importance in understanding the mechanisms of tic disorders, and potentially of other disorders of compulsion. Moreover, their findings lend strength to a growing body of work demonstrating potential stimulation sites for treating tic disorders with non-invasive stimulation.

In general, their science is plausible and was executed using well-established methods. However, there are numerous areas with spelling and phrasing errors that would be obvious to a senior author. This suggests that the senior authors of this manuscript did not read it, a serious concern. Underscoring this point, the interpretation of prior network analyses in tic disorders is rather facile, and elides considerable differences between studies that must be considered in interpreting their data. Again, this sort of writing suggests that the senior authors did not weigh in. I am not concerned about typographic errors or oversights in citation/figure generation for their own sake – rather, the persistence of these errors suggests that this otherwise impressive study has not been thoroughly vetted by the team of authors. To be publishable, the senior authors from each center must actually read the full paper and add perspective to the Introduction and Discussion. If the response is, "Yes, we did that..." then I suggest they go back again. This paper needs more nuance and perspective to shape its presentation and conclusions. The paper will be substantially strengthened by a tough read by its own authors.

Abstract:

You stated, "Gilles de la Tourette Syndrome (GTS) is the most severe form of chronic tic disorders, characterized by uncontrollable motor actions and vocalizations."

- A GTS diagnosis does not imply or rely upon tic severity. I have many patients with Chronic Motor Tic Disorder whose symptoms are more severe than many patients with GTS.
- Tics have a degree of suppressibility, by definition. They are not uncontrollable.

Introduction:

- As above – tics are not uncontrollable, and patients with GTS are neither the most severe or the most chronic patients with tics. However, it is certainly true that patients whose tics warrant DBS are the most severe, and are most commonly afflicted by GTS.
- Note typo: prevalent in
- Note repetition, line 122: "Here, we empirically investigate this relationship empirically:"
- It is surprising that your Introduction does not discuss lateralization of function in impulse control and tic disorders.

Methods:

- The YGTSS is reliable over repeated measurements and is a good match for subjective tic severity. But it also varies with normal variation in GTS severity independent of treatment. That is, the YGTSS follows the natural ebbs and flows of tic disorders, so averaged measures over a short time interval are more accurate than single measures. Please detail how you used the YGTSS, if these were single measures or averages, and how you accounted for normal, baseline variance in tic severity vs. treatment effects.

- Please supply the resolution of the T1 and functional scans utilized. The dimensions of several of the structures of interest include only single digit numbers of voxels at standard fMRI resolution (eg, CM). Imaging such small structures is

challenging, and your readers deserve to know more about how you achieved and confirmed the accuracy of your anatomic segmentations.

- Drs. Benjamini and Hochberg deserve to be cited: Controlling the False Discovery Rate: A Practical and Powerful Approach to Multiple Testing.

- line 239: "we avoided a circular analysis that would be overly similar to the analysis of thalamic stimulation sites." Why not use a whole hemisphere mask that excludes the thalamus and ingressing white matter? By leaving out all subcortical structures, you are almost certainly leaving helpful targets out of your map.

- line 273: "all other networks" could mean so many things. Please specify the number and which networks. Note: this is the type of superficial statement that would have been corrected if senior authors had read the manuscript.

Results:

- Line 277: "mean age: 32.4 years \pm 10.8 standard deviations"
Did your age distribution truly vary by 11 SD? I'm not even sure what this means.

- Line 281: "were stimulated by 85%" Do you mean "stimulated IN 85%"?

- Figure 2B: Are these left and right hemispheres (upper and lower panels)? Explain more what you're indicating by "CON Overlay" and "SCAN Overlay". Same with Fig. 3B and Fig. 4.

- Figures 2B and 3B: Your correlations with SCAN sites are indeed suggestive of correlation with tic-controlling networks. But large parts of the SCAN sites are uninvolved. Please quantify the % of SCAN Overlay voxels that have vs. do not have significant functional connectivity, and compare this with other tested nodes. How large and specific is this overlap with SCAN?

- Line 311: Your sensorimotor network finding verges on significance. Can you do a post-hoc comparison to learn which parts of SMN are driving this correlation?

- Figure 3a, Spearman r section: please explain this finding in greater depth in your figure caption. I'm not sure what this is supposed to tell me.

- Line 341: Your middle and inferior SCAN sites have considerable overlap with the Tic Reduction Heat map, but the superior SCAN site appears to have little correlation. Please measure this difference and elaborate on what connectivity with different parts of SCAN might indicate.

Discussion:

- Figure 6: the purple voxels are hard to make out on the template brain background. Overlap appears likely, but you should not leave this as an assumption – show us where they overlap.

- Line 412: "can potentially to uncover"

- Lines 450-451: The insula is highly diverse, and it is not sufficient to name it as a solitary structure. How does the intra-insular location of your findings compare with the intra-insular location of these prior studies?

Line 491: typo "neural activity data o "

(Remarks on code availability)

Reviewer #3

(Remarks to the Author)

Considering that different targeting techniques, even when aimed at the same target, may vary significantly across centers, and given that the hypothetical changes in network connectivity resulted in differential outcomes in tic reduction, were these clusters (e.g., CON or SCAN) correlated with different target locations within the medial thalamic CM, Voi, or subregions? Could your connectivity analysis identify a preferable hotspot in general or clinical subtypes?

Although TS is typically described in a consistent clinical manner, it is well known that subtypes of tic prevalence or behavioral comorbidity can vary widely. Were there any consistent findings in network connectivity, based on preoperative resting-state or similar data, which correlated with specific tic subtypes or comorbidities? If so, were there any specific

patterns that could predict or correlate with the best response to thalamic DBS?

If not, considering that functional imaging could data could well correlate with TS clinical presentation as it has been shown with tremor predominant or akinesia predominant PD, is there a point in using data from normal subjects to predict networks involved in pathological events and their neural correlates?

Given that the insular region was highly involved in the positive correlation findings, could you provide higher resolution images focusing on the claustrum, a hypothetically involved network hub for consciousness that integrates associative cortical and basal ganglia networks? Could the claustrum also have been involved in the tic-producing network?

(Remarks on code availability)

Version 1:

Reviewer comments:

Reviewer #1

(Remarks to the Author)

The authors have done an admirable job responding to my comments/suggestions

(Remarks on code availability)

Reviewer #2

(Remarks to the Author)

The authors have substantially improved the clarity of the manuscript and provided several new insights as a result of these revisions. I am satisfied that these changes meet the standard for plausibility and reproducibility. However, there are three wording changes that will improve the clarity of their message.

You wrote "p. 8: "In the main analysis, we included 37 patients with severe GTS (mean age: 32.4 years \pm 10.8 standard deviations; range: 18 to 65; 5 females)."

You are writing this incorrectly, and the solution was not to add the range of values. Your standard deviation = 10.8 years. It is NOT 10.8 standard deviations. Instead, write "32.4 years \pm 10.8 years (standard deviation);"

You wrote "p. 10: "... executive central 6% and 0.57; ..." Earlier in the paragraph you identify this network as central executive, but reverse it here.

You wrote "p 14: "... with fewer tics involving the feet and body core⁴⁹. Similar, the premonitory urge to tic..." You mean "Similarly" rather than "Similar".

Fix these three wording errors and I'm satisfied. No need to re-review.

Nice job!

(Remarks on code availability)

We would like to thank the Reviewers for their assessment of our work and the constructive feedback. In the following, we will answer their comments and critiques point by point. We have numbered the comments by the reviewers for readability. The reviewer comments are displayed in green. Author answers are displayed in blue, with respective changes in the manuscript are displayed in red.

REVIEWER COMMENTS

Reviewer 1:

1. In this manuscript, “A critical role for Action-Related Functional Networks in Gilles de la Tourette Syndrome”, the authors test the hypothesis that functional connectivity to action-related networks, investigated in a multicenter cohort study of patients undergoing thalamic DBS or having tic-inducing spontaneous lesions, would be associated with tic reduction. The principle findings in the study are: 1) Greater tic reduction in patients receiving thalamic DBS is linked to higher functional connectivity of the DBS sites to action related networks (as measured by using the patient sites to quantify connectivity in a openly available dataset of 1000 healthy adults). 2) The findings were replicable in a separate, smaller dataset of DBS receiving patients, and 3) Tic-inducing lesions revealed a comparable network connectivity profile, implicating the same action related networks. The authors have produced a well-written and (mostly) well-figured manuscript that addresses a deeply interesting hypothesis and takes full advantage of collaboratively, multicenter collected patient data as well as a fairly large public MRI dataset collected from healthy individuals. In addition, the authors have capitalized promptly on very recent advances from precision functional mapping using resting state functional connectivity MRI that have reshaped in a quite remarkable way how we think about the representation of the somatic motor system in primary motor cortex. The so-called Somatic Cognitive Action Network constitutes a reframing of how primary motor cortex is organized. An important takeaway from this work is that the reframing of the classic motor homunculus has implications that go beyond updating the schematic renderings of the motor homunculus dating back over a century -- the organizational and functional features of the SCAN very likely have clinical significance. That insight would be impossible to attain without the emergence of deeper understanding of the organization of the brain’s functional network architecture through cutting edge advanced neuroimaging and network informed analyses. The methodology is clearly presented and sound, and is sufficiently detailed for others to reproduce the approach. This paper, therefore, has broad implications beyond the relatively narrow scope of a pediatric onset movement disorder (though the implications for Tourette syndrome are certainly substantial). The work is exciting, thought provoking, and important.

We appreciate the reviewer’s overall positive feedback.

2. Minor criticisms: The paper, while crisply written, would benefit from figures and figure labeling more mindful of the non-expert reader. For example, a figure showing brain surface renderings of each of the functional networks examined would be very helpful. Such a figure would also help make the SCAN and CON overlays more understandable to the uninitiated. Figure 6 provides A, P, R, L labels (not mentioned in the legend), but the other figures do not provide such orienting information. Overall, the

recommendation is to reexamine and edit the figures taking into account the broader readership of this journal.

We agree that the figures have potential for improvement. We followed the reviewer's suggestion and revised all figures to ensure that readers outside the neuroimaging/neuromodulation community may follow the content. To this end, we have added more detailed descriptions about hemispheres and orientation of figures and also tried to avoid abbreviations in the figures (see revised figures). As recommended, we also added a figure displaying the SCAN and CON networks in the method figure and we adapted the descriptions for better understanding as detailed below:

p. 4, Figure 1:

Figure 1: Method Overview. *a) As regions of interest, we chose functional networks based on resting-state connectivity known to be involved during actions in humans. The cingulo-opercular network (CON), also referred to as action-mode network, is linked to the processing of arousal, error detection and pain sensation and thus states that call for action. The somato-cognitive action network (SCAN) has recently been described as a network relevant for action planning and complex body movements. Both networks show high connectivity with each other. b) For each subject in a cohort of patients with GTS undergoing thalamic DBS for tic reduction (N = 37), electrodes and the volume of activated tissue (i.e., stimulation site) were reconstructed in standard space. Using a publicly available resting-state functional connectome acquired in healthy participants (N = 1,000), we computed the functional connectivity of each bilateral pair of stimulation sites with all other brain voxels. The resulting connectivity maps were then used for group analysis. c) Initially (1.), a map of voxel-wise average connectivity was computed (termed average map). Subsequently (2.), the average connectivity in a priori defined regions of interest, i.e., CON and SCAN, was correlated with tic reduction post-DBS. In an alternative data-driven whole-brain approach (3.), we calculated a map of voxel-wise significant correlations (pFDR < 0.05) between connectivity and tic reduction (termed R-map), signifying an optimal connectivity pattern. Lastly (4.), the weighted connectivity of this R-map, limited to cortical voxels, to each other brain voxel was computed. This resulted in a map where each voxel contained the likelihood of matching the optimal connectivity profile, suggesting potential novel cortical target networks (termed target heat map).*

p. 11, Figure 3 (exemplary):

3. The limitations section does include some language about the benefit of precision functional imaging approaches and how it is untenable at this time to endeavor to highly sample patients with such severe

ticcing. That said, the manuscript would benefit from a deeper dive into the topic. For example, some proportion of the variance assuredly comes from the fact that the presumptive functional connectivity of individual patient's electrode sites is being mapped to a population sample that is non highly sampled and placed in standard space. Again, this comment is not intended to be a criticism. Rather, the reader would benefit from understanding the potential consequences.

The issue of using a normative connectome that is neither patient- nor disease-specific is a major limitation. We have now extended the limitation section substantially in order to address this shortcoming as below:

p. 17: "Further, the current study used a normative functional connectome that was derived from 1,000 healthy participants, thereby neglecting potential disease-specific connectome disruptions or individual differences in connectivity. This approach may serve as a broader characterization of the underlying stimulated or lesioned networks in an averaged, healthy brain, where the normative connectome can be understood as an atlas of average brain connectivity. This normative method has previously effectively predicted DBS treatment responses across various conditions^{38,39,44}. Beyond DBS, neural networks of numerous neuropsychiatric symptoms caused by heterogeneous focal brain lesions could be explained using the same fMRI data^{43,71-74}. Our investigation of lesions causing tic symptomatology matched the connectivity profile derived from DBS, indicating promising generalizability of the tic reduction map across brain interventions despite the use of normative connectivity data. Nonetheless, patient- or disease-specific information about brain connectivity may allow to explain more variance in DBS outcomes. This is particularly of interest since GTS constitutes a neurodevelopmental disorder. Thus, it could be possible that in people with GTS, the investigated networks develop differently across the lifespan. Indeed, an earlier study suggests that the CON shows altered maturation in adolescents with GTS compared to age-matched healthy controls⁷⁵. A similar conclusion, i.e. altered maturation of functional brain networks in GTS, was drawn from a more recent study. Here, a machine learning-derived classification of GTS using functional connectivity networks successfully distinguished patients with GTS from healthy controls – however, the distinguishing features differed between children and adults, pointing towards different neurodevelopmental trajectories of functional brain networks in children and adults with GTS⁷⁶. That being said, the principal functional brain networks were well traceable in the patient cohorts in both studies. Thus, although quantitative differences in these networks may be assumed, it appears unlikely that the general architecture of functional brain networks is drastically different in patients with GTS."

4. Along these lines, while precision mapping may not be tenable in patients with severe ticcing, it likely is tenable in patients with mild to moderate tics. The question of whether network architecture is disrupted developmentally in the context of Tourette syndrome, signal may be evident in such precision mapped individuals. Not a request for additional investigation.

We agree that such a connectome could be built. We added this potential outlook in the paragraph as follows:

p. 17: "Along these lines, a study of patients undergoing DBS for Parkinson's disease showed that the usage of patient-specific connectivity was not significantly superior to normative connectivity data and that both approaches revealed similar optimal brain networks to be targeted⁷⁷. Nonetheless, given the potential disease-specific brain alterations, we assume that at least some more variance of outcomes may be explainable by using disease-specific or even patient-specific connectivity data. However, obtaining high-quality resting-state fMRI data from unmedicated, severely affected patients with tic disorders that may qualify for DBS is hardly feasible, meaning that this approach is currently the only way to investigate functional connectivity estimates in patients with tic disorders who received DBS in

a larger sample. One potential solution for this limitation could be to investigate a disease-specific functional group connectome derived from patients with less severe tics, where rs-fMRI of sufficient quality can be assessed. With such a group connectome, it could be tested whether predictability of outcomes could be improved compared to the currently used normative group connectome.”

5. The term “hub” is used multiple times in the paper. It is not clear that the term is used accurately with respect to its meaning in networks. Rather, it seems that “hub” is being used to mean “target”.

Indeed, the term “hub” has not been used in the adequate meaning of network neuroscience. We thus have replaced it throughout the manuscript.

Reviewer #1 (Remarks on code availability):

To my understanding, there is sufficient code information for replication.

Reviewer 2:

1. In this article, Baldermann and colleagues have described two cohorts of patients with Tourette syndrome who underwent DBS to reduce tic severity, and a cohort of patients with lesion-induced tics. The research team assessed resting state functional connectivity at DBS sites that were more vs. less effective at tic reduction and demonstrated that stimulation sites that were effective at reducing tics had significant functional connectivity with the CON and SCAN networks. They replicated these findings in previously-published cohorts of patients treated with DBS for tics and in patients whose brain lesions induced tics, demonstrating similar functional network correlations. These findings are of considerable importance in understanding the mechanisms of tic disorders, and potentially of other disorders of compulsion. Moreover, their findings lend strength to a growing body of work demonstrating potential stimulation sites for treating tic disorders with non-invasive stimulation. In general, their science is plausible and was executed using well-established methods. However, there are numerous areas with spelling and phrasing errors that would be obvious to a senior author. This suggests that the senior authors of this manuscript did not read it, a serious concern. Underscoring this point, the interpretation of prior network analyses in tic disorders is rather facile, and elides considerable differences between studies that must be considered in interpreting their data. Again, this sort of writing suggests that the senior authors did not weigh in. I am not concerned about typographic errors or oversights in citation/figure generation for their own sake – rather, the persistence of these errors suggests that this otherwise impressive study has not been thoroughly vetted by the team of authors. To be publishable, the senior authors from each center must actually read the full paper and add perspective to the Introduction and Discussion. If the response is, “Yes, we did that...” then I suggest they go back again. This paper needs more nuance and perspective to shape its presentation and conclusions. The paper will be substantially strengthened by a tough read by its own authors.

We thank the reviewer for the valuable feedback. We genuinely appreciate the critique and the hard work the reviewer put in to help us further improve our manuscript. We specifically made sure that all senior authors re-read the manuscript with particular focus on the intro and discussion sections and believe that we were able to substantially revise these sections and the manuscript, at large.

2. Abstract: You stated, “Gilles de la Tourette Syndrome (GTS) is the most severe form of chronic tic disorders, characterized by uncontrollable motor actions and vocalizations.”
- A GTS diagnosis does not imply or rely upon tic severity. I have many patients with Chronic Motor Tic Disorder whose symptoms are more severe than many patients with GTS.
- Tics have a degree of suppressibility, by definition. They are not uncontrollable.

We thank the reviewer for these clarifications. Indeed, the statement that GTS would generally constitute the most severe form of tic disorders is not sufficiently accurate. With this statement, we referred to the paper by Müller-Vahl et al. (Eur Child Adolesc Psychiatry. 2019; doi: 10.1007/s00787-018-01272-7), where patients with GTS had *on average* more severe tics than chronic motor tic disorders. But, as the reviewer rightfully stated, some patients with chronic motor tics may show more severe tics than some patients with GTS. Therefore, we have now omitted this statement.

Regarding the term 'uncontrollable', we aimed to describe the phenomenology where patients can suppress their tics, but not sufficiently or consistently enough to be tic-free. We chose the word uncontrollable to describe this unique character of tic disorders, which we also discussed in the introduction and in the discussion. However, we agree that our wording might not capture the phenomenology well enough. Thus, we now changed the adjective to “unwanted”, bearing in mind the discussion about volitional control over tics (and thus avoiding the term “involuntary”).

3. Introduction: As above – tics are not uncontrollable, and patients with GTS are neither the most severe or the most chronic patients with tics. However, it is certainly true that patients whose tics warrant DBS are the most severe, and are most commonly afflicted by GTS.
- Note typo: prevalentin

- Note repetition, line 122: “Here, we empirically investigate this relationship empirically:”

- It is surprising that your Introduction does not discuss lateralization of function in impulse control and tic disorders.

We agree and as outlined above now omitted the term “uncontrollable”. We also corrected the typo (“prevalentin” to “prevalent in”) and rephrased the repetition. We apologize for this negligence.

Regarding the lateralization of function in impulse control and tic disorders we assume that the reviewer refers to the well-established observation that impulse control in the sense of behavioral inhibition is usually lateralized to the right brain hemisphere, particularly the right inferior frontal gyrus. Regarding lateralization in tic disorders, the literature appears to be less definite. Especially those studies investigating brain activity during tics or urges report bilateral activation of the key brain areas, i.e. the insula/operculum, medial prefrontal cortex and parietal cortex (e.g. Bohlhalter et al., Brain 2006; Neuner et al., Front Hum Neurosci 2014). On the other hand, case control studies report a right-hemispherical dominance of these networks when compared to healthy controls.

We believe that the aspect of behavioral inhibition or impulse control is outside the scope of our article since we cannot state that DBS in our patients changes inhibitory control. However, we now acknowledge that a right-hemispherical dominance of tic/urge-related network when compared to healthy controls has been reported in the literature:

p. 3: “These networks activate on both hemispheres before and during tic executions^{9,10}, although different case-control studies suggest a right-hemispheric functional dominance in GTS compared to healthy controls^{11,12}”.

4. Methods: The YGTSS is reliable over repeated measurements and is a good match for subjective tic severity. But it also varies with normal variation in GTS severity independent of treatment. That is, the YGTSS follows the natural ebbs and flows of tic disorders, so averaged measures over a short time interval are more accurate than single measures. Please detail how you used the YGTSS, if these were single measures or averages, and how you accounted for normal, baseline variance in tic severity vs. treatment effects.

We used a single assessment of the YGTSS, which captures the tic severity over the course of one week. However, we acknowledge that the natural waxing and waning of symptoms in tic disorders introduces variability in the data. Unfortunately, only one measurement was available across all subjects, which could be directly linked to a specific stimulation setting. In subsequent visits, the stimulation parameters often changed, making it impossible to use multiple YGTSS assessments consistently across the cohort.

While we recognize that this approach may lead to an over- or underestimation of treatment effects in individual patients, the relatively large sample size in our study mitigates the potential for general distortion in the overall analysis. Moreover, by doing so, we follow the approach used in almost any DBS network mapping paper we are aware of (for a review e.g. see Horn & Fox 2020 NeuroImage).

We have now clarified this in the manuscript and acknowledge this limitation. Specifically, we selected the earliest single assessment within a six to twelve-month post-intervention period as the follow-up measure. We have added the following explanation to the limitations section:

p. 5: “As follow up, the earliest single assessment in a period between six to twelve months post-intervention was chosen.”

p. 18: “Further, the treatment effects of DBS were assessed by calculating the change in tic severity from single pre- to postoperative assessment of the YGTSS, which captures individual tic severity over one week. Since the severity of tics naturally waxes and wanes over time, multiple assessments would have been advantageous to account for baseline variance in tic severity. However, as stimulation settings were subject to changes across clinical visits, only one assessment per stimulation setting was available across the cohort. Although this may introduce some variability on an individual level, the relatively large sample size in our study helps to minimize potential distortions in the overall findings.”

5. Please supply the resolution of the T1 and functional scans utilized. The dimensions of several of the structures of interest include only single digit numbers of voxels at standard fMRI resolution (eg, CM). Imaging such small structures is challenging, and your readers deserve to know more about how you achieved and confirmed the accuracy of your anatomic segmentations.

For the reconstruction of electrodes in standard space, we used clinical T1 imaging and postoperative MRI or CT scans that were originally employed to plan and control the procedure. Since these scans were obtained at different centers, the exact imaging parameters varied. We have now provided an overview of the individual resolutions in Supplemental Table 2. On average, the dimensions for the preoperative imaging were $0.6 \times 0.6 \times 1.1$ mm, and for the postoperative imaging, they were $0.6 \times 0.6 \times 1.4$ mm.

Regarding the functional connectome, the resolution was $3 \times 3 \times 3$ mm (Yeo et al. 2011, J Neurophysiol). We want to emphasize that our experience is that averaging 1,000 scans typically leads to stronger detection of smaller signals, especially in the subcortex. For instance, the CM has been described in all

datasets described by the original authors of the SCAN network (Gordon et al. 2023 Nature) which included multiple datasets of comparable resolution. For the functional connectivity estimates, we added the following description to the manuscript:

p. 6: “Functional images were acquired using a 3 Tesla Siemens scanner with a resolution of $3 \times 3 \times 3$ mm. Preprocessing of the blood oxygen level-dependent time courses included regression of global signal, white matter, and cerebrospinal fluid signals, as well as the six motion parameters. Smoothing was performed using a 6 mm full-width at half-maximum (FWHM) Gaussian kernel, as previously described³⁷.”

6. Drs. Benjamini and Hochberg deserve to be cited: Controlling the False Discovery Rate: A Practical and Powerful Approach to Multiple Testing.

Thank you, this was added as suggested.

7. line 239: “we avoided a circular analysis that would be overly similar to the analysis of thalamic stimulation sites.” Why not use a whole hemisphere mask that excludes the thalamus and ingressing white matter? By leaving out all subcortical structures, you are almost certainly leaving helpful targets out of your map.

The issue of subcortical connectivity within the tic reduction R-map indeed requires further clarification. Our approach focused on the cortical connectivity profile, deliberately excluding subcortical structures from the map to prevent overlapping with regions that might be directly influenced by DBS interventions. This decision was made to ensure that the analysis of cortical regions would complement the R-map, which primarily utilizes subcortical information and follows the approach introduced previously (Li et al., Biological Psychiatry 2021).

However, we acknowledge that this approach might exclude potentially valuable information regarding regions outside the thalamus that contribute to tic reductions through their connections with the target area. To address this, we have now chosen to display the subcortical clusters of the Tic Reduction R-map in detail (Figure 3e). This reveals that no clusters within the thalamus survive the significance threshold, with only very small clusters in the posterior part of the putamen remaining significant. Therefore, masking only the thalamus, as opposed to the entire subcortex, results in almost identical findings.

To provide a comprehensive view, we have now included both approaches and added the resulting map in Supplemental Figure 2. The following changes have been made:

p. 7: When using the R-map as seed for this analysis, we masked the subcortex to avoid a circular analysis that would be overly similar to the analysis of thalamic stimulation sites. Notably, masking only the thalamus led to highly similar results (see Supplemental Figure 3).

p. 11, Fig. 3e:

Insular and Subcortical Clusters of Tic-Reduction R-Map

Supplement, p. 1:

Supplemental Figure 2: Tic Reduction Heat Map, derived from thalamus-masked R-Map. Displayed is the heat map that results from the Tic-Reduction R-map, when only the thalamus is masked instead of the whole subcortex (see Figure 4). Both maps are highly similar, since the original R-map included rarely any voxels in the subcortex that resided outside the thalamus.

8. line 273: “all other networks” could mean so many things. Please specify the number and which networks. Note: this is the type of superficial statement that would have been corrected if senior authors had read the manuscript.

In this sentence, we referred to the previous description of the networks employed in the DBS-specific analysis. We have now clarified the networks as follows:

p. 8: “... between CON and SCAN and further major functional resting-state networks (i.e. sensorimotor, central executive, visual, language and default mode network according to CAREN⁴² atlas) using an $p_{FDR} = 0.05$ to control for multiple comparisons”.

9. Results: Line 277: “mean age: 32.4 years \pm 10.8 standard deviations”
Did your age distribution truly vary by 11 SD? I’m not even sure what this means.

The standard deviation is indeed 10.8 years. We now also added the age range to make the data more comprehensible.

p. 8: “In the main analysis, we included 37 patients with severe GTS (mean age: 32.4 years \pm 10.8 standard deviations; range: 18 to 65; 5 females).”

10. Line 281: “were stimulated by 85%” Do you mean “stimulated IN 85%”?

Thank you. We changed this sentence reporting “stimulated in 85%”.

11. Figure 2B: Are these left and right hemispheres (upper and lower panels)? Explain more what you’re indicating by “CON Overlay” and “SCAN Overlay”. Same with Fig. 3B and Fig. 4.

We agree that the figures can be improved to be self-explanatory. Also following the comments made by reviewer 1, we now revised all figures by adding the respective hemispheres, orientation and avoiding abbreviations when not necessary. By doing so, we also clarified that the wording “CON overlay” was meant to indicate that the maps are shown with the additional outlines of the CON or SCAN networks:

e.g. Fig. 2:

12. Figures 2B and 3B: Your correlations with SCAN sites are indeed suggestive of correlation with tic-controlling networks. But large parts of the SCAN sites are uninvolved. Please quantify the % of SCAN Overlay voxels that have vs. do not have significant functional connectivity, and compare this with other tested nodes. How large and specific is this overlap with SCAN?

We agree that quantifying the overlap between the tic reduction map and the SCAN sites is crucial to accurately assess the specificity and extent of the correlation. To address the reviewer’s request, we now did so, and we also quantified overlaps with other relevant network maps. In addition to calculating the percentage of overlap, we also measured the balanced accuracy to provide a comprehensive assessment of the connectivity specificity and sensitivity (see below for a detailed description). This analysis allowed us to not only quantify the extent of overlap but also to assess how specific and sensitive the connectivity of the SCAN and other networks is in relation to tic reduction. The results, including the balanced accuracy values, support our conclusion that the cingulo-opercular

network (CON) showed the highest overlap with the tic reduction R-map, followed by the somato-cognitive action network (SCAN).

p. 7: “We quantified the overlap of the R-map indicative of tic reduction with our ROI networks CON and SCAN as well as with the previously mentioned functional brain networks derived from the CAREN atlas⁴² (sensorimotor, central executive, visual, language and default mode network). Additionally, we calculated the balanced accuracy as the mean of sensitivity and specificity, offering a balanced evaluation of true positive and true negative rates. Sensitivity (True Positive Rate) was calculated as the ratio of overlapping voxels (voxels in the positive R-map that also belong to the network) to the total number of voxels in the network (overlapping + non-overlapping voxels in the network). Specificity (True Negative Rate) was calculated as the ratio of non-overlapping voxels outside the network (voxels not in the positive R-map and not in the network) to the total number of non-network voxels (non-overlapping + overlapping voxels outside the network). Balanced accuracy was then determined by averaging the sensitivity and specificity, providing a robust metric that accounts for both the true positive and true negative rates.”

p. 10: “Quantitatively, the R-map associated with greater tic reduction showed the highest overlap and highest balanced accuracy with the CON (overlap 46% and balanced accuracy 0.89), followed by the SCAN (26% and 0.73). All other networks showed lower overlaps and balanced accuracies (sensorimotor 13% and 0.60; visual 7% and 0.55; executive central 6% and 0.57; language < 1% and 0.50; default mode < 1 % and 0.48).”

13. - Line 311: Your sensorimotor network finding verges on significance. Can you do a post-hoc comparison to learn which parts of SMN are driving this correlation?

We thank the reviewer for this comment as it has prompted us to conduct further analysis of the data. Indeed, the sensorimotor network (SMN) findings were close to reaching significance. It's important to note that the SMN in the CAREN atlas also includes inter-effector regions of the SCAN network and small parts of the CON. To eliminate any potential ambiguity in our findings, we decided to mask out all voxels that are part of the CON and SCAN from the SMN. After doing so, the pFDR value for the SMN changed to 0.063, which still indicates a trending significance.

To further address the reviewer's question, we conducted a post-hoc analysis focusing on the precentral and postcentral gyri, which are the primary anatomical regions within the SMN. Our analysis showed that both regions had significant correlations with tic reduction, although these results were not corrected for multiple comparisons (precentral gyrus: $r = 0.44$, $p_{\text{uncorrected}} = 0.003$; postcentral gyrus: $r = 0.34$, $p_{\text{uncorrected}} = 0.023$). Upon examining the scatter plots, we observed that all patients exhibited negative connectivity values for the postcentral gyrus, and 92% showed negative connectivity values for the precentral gyrus. This indicates that the correlations are driven by negative values, suggesting that patients who responded well to DBS showed reduced or no anticorrelation with the sensorimotor cortex, whereas patients who did not respond as well showed stronger anticorrelations.

This pattern suggests that the mechanism underlying the SMN's role may differ from that of the CON and SCAN, where higher positive correlations with DBS outcomes indicate better responses. Essentially, it would imply that stimulation sites of top responders would show high connectivity with CON and SCAN alongside no correlation with the SMN. As requested by the reviewer we now added the following explanation of the post-hoc analysis:

p. 10: “Since the sensorimotor network showed a trend towards significance, we performed a post-hoc analysis to investigate which parts of this network drove the effect. Both the precentral and

postcentral gyri showed positive correlations with outcomes. However, unlike the positive correlations with the CON and SCAN, all patients showed negative connectivity values for the postcentral gyrus, and 92% showed negative connectivity values for the precentral gyrus. Thus, the association of tic reduction with connectivity to the sensorimotor network likely reflects a different mechanism, where reduced or no anticorrelation with the network correlates with better outcomes after DBS. Specifically, this analysis indicates that stimulation sites of top-responders would likely exhibit strong connectivity with the CON and SCAN, while showing no connectivity or only low negative connectivity with the SMN.”

14. Figure 3a, Spearman r section: please explain this finding in greater depth in your figure caption. I’m not sure what this is supposed to tell me.

We have now expanded the figure caption and hope that the figure is now more self-explanatory:

p. 11: “a) In a region-of-interest (ROI) analysis, we correlated the percentage tic reduction after DBS with the respective connectivity of stimulation sites with the cingulo-opercular (CON) (top left) and somato-cognitive action network (SCAN) (top right). Non-parametric permutation testing (at the bottom) revealed a significant positive relationship between tic reduction and connectivity with the CON ($r = 0.62$, $p < 0.001$) and SCAN ($r = 0.47$; $p = 0.002$), indicating that greater connectivity between the DBS sites and these networks is associated with more substantial reductions in tic severity.”

15. Line 341: Your middle and inferior SCAN sites have considerable overlap with the Tic Reduction Heat map, but the superior SCAN site appears to have little correlation. Please measure this difference and elaborate on what connectivity with different parts of SCAN might indicate.

We indeed observed that connectivity with the middle and inferior inter-effector regions of the SCAN showed a stronger relationship with treatment outcomes compared to the superior inter-effector regions. Following the reviewer’s suggestion, we quantified these associations. Our analysis revealed strong and significant correlations ($r > 0.5$; $p < 0.001$) for the inferior and middle inter-effector regions, while the superior inter-effector regions exhibited a weaker, statistically trending, but not significant correlation ($r = 0.276$, $p = 0.052$). Although these findings are speculative at this stage, we discuss them in the context of a potential somatotopic arrangement of the inter-effector regions and the distribution of tics across the body.

p. 7: “Finally, we conducted a post-hoc analysis to investigate how the three different inter-effector regions (inferior, middle, and superior) contributed to the treatment outcomes. We calculated the connectivity of DBS sites to each inter-effector region and separately correlated these estimates with the percentage reduction in tic severity.”

p. 9f: “A post-hoc analysis revealed that the positive association of tic reduction with connectivity to the SCAN was mainly driven by the inferior and middle inter-effector regions, both showing strong significant correlations ($r = 0.596$; $p < 0.001$ and $r = 0.513$; $p < 0.001$). In contrast, the superior inter-effector regions showed a weak correlation that approaches but does not reach statistical significance ($r = 0.276$; $p = 0.052$) (Figure 3c).”

p. 14: “Within the SCAN, a post-hoc analysis showed that the inferior and middle inter-effector regions predominantly contributed to the overall positive association of tic reduction and functional connectivity of stimulation sites. While the exact roles of the different inter-effector regions are not

fully understood¹³, their involvement may reflect the somatotopic organization of the sensorimotor cortex. This organization would align with the distribution of tics in GTS, where most tics involve the head and face muscles, followed by the hand/shoulder muscles, with fewer tics involving the feet and body core⁴⁹. Similar, the premonitory urge to tic is most commonly experienced in the head/face region, followed by the upper extremity, and is least experienced in the lower extremities⁵⁰. Therefore, although speculative at this moment, the differential engagement of the inferior, middle, and superior inter-effector regions observed in our tic reduction R-map might reflect the somatotopic distribution of tics and premonitory urges.”

16. Discussion: - Figure 6: the purple voxels are hard to make out on the template brain background. Overlap appears likely, but you should not leave this as an assumption – show us where they overlap.

We agree, our figure does not capture the question of overlap. Thus, we have now decided to show the outline of the cingulo-opercular network to ensure that readers may fully see where the maps overlap.

17. Line 412: “can potentially to uncover”

We corrected the typo.

18. Lines 450-451: The insula is highly diverse, and it is not sufficient to name it as a solitary structure. How does the intra-insular location of your findings compare with the intra-insular location of these prior studies?

We have now further specified the extension of the insular clusters. In the tic reduction R-map, the clusters have been located in the posterior insula, extending to the middle and anterior part of the insula. Further, it encompasses the operculum and subcortically the posterior part of the claustrum. In the target heat map, the peak cluster (with $R > 0.6$) is located in the bilateral dorsal anterior insula, reaching the rolandic operculum. This matches the findings by Tinaz et al., who reported a relative anterior-shift of insular connectivity within the urge-tic network, represented by increased connectivity of the anterior insula with the SMA. We clarified this now in the results and discussion section:

p. 10: “The insular cluster encompassed the posterior short and long gyrus, as well as the middle and anterior short gyrus and the rolandic operculum.”

p. 12: “Within this map, peak clusters (containing voxels with $R > 0.6$) were located in the bilateral supramarginal gyrus, rolandic operculum/anterior dorsal Insula, and the SMA (Figure 6a), all located within the CON (Supplemental Table 1)”.

p. 16: “The anterior dorsal insula was the third prominent region of our tic-reduction target heat map. fMRI studies have shown that the anterior insula is hyperconnected with the SMA in patients with tic disorders and that this connectivity correlates with urge severity¹¹.”

19. Line 491: typo o “

We corrected the typo and we thank the reviewer for catching this oversight.

Reviewer 3:

1. Considering that different targeting techniques, even when aimed at the same target, may vary significantly across centers, and given that the hypothetical changes in network connectivity resulted in differential outcomes in tic reduction, were these clusters (e.g., CON or SCAN) correlated with different target locations within the medial thalamic CM, Voi, or subregions? Could your connectivity analysis identify a preferable hotspot in general or clinical subtypes?

The question regarding the optimal target region within the thalamus is indeed highly interesting and relevant. In response to the reviewer's remark, we now show the peak spot of the target heat map within the thalamus (Figure 4b). This analysis indicates that the heat map's highest connectivity values are located in the anterior and superior part of the centromedian (CM) nucleus. Thus, this cluster may represent a potential hotspot derived from our analysis. Additionally, we demonstrate that both the CON and the SCAN exhibit similar connectivity profiles in the thalamus, reinforcing our conclusion that these networks are crucial for treatment outcomes.

However, we wish to emphasize that a hotspot derived from functional connectivity analysis should be interpreted with caution. The resolution of the functional connectome used in this study is 3x3x3 mm, which is a common resolution of resting-state fMRI data. While this resolution is adequate for using stimulation sites as seeds and exploring whole-brain functional networks including subcortical contributions—since the resulting volume of activated brain tissue is larger than the voxel size—there are inherent limitations when attempting to pinpoint precise stimulation targets within the thalamus.

To address these limitations, an alternative or complementary approach could involve voxel-wise analyses of the electric field or the volume of activated tissue, which might generate probabilistic stimulation maps predictive of therapeutic outcomes. In fact, two previous studies utilizing such probabilistic maps identified the anterior dorsal CM nucleus as an optimal target, aligning with our findings^{27,61}. However, an earlier analysis did not reveal a consistent pattern, likely due to extended and heterogeneous follow-up periods in outcome assessments⁶².

Therefore, further research involving larger samples and well-controlled outcome assessments is necessary to accurately delineate the optimal thalamic target for deep brain stimulation. Additionally, it would be valuable to explore whether similar functional networks identified in our current analysis are also critical for DBS targeting the globus pallidus internus.

p. 12, Fig. 4b:

Figure 2: Target heat map for Tic Reduction. a) We computed a target heat map containing the voxel-wise connectivity to the R-map, to identify a network profile that match the optimal connectivity profile derived from thalamic stimulation. The highest matches were observed within the cingulo-opercular network (CON). Within the motor cortex, the somato-cognitive action network (SCAN) showed the highest similarity of connectivity with the tic-reduction R-map. b) Within the thalamus, the heat map peaked in the superior and anterior part of the centromedian nucleus (outlined in white). Connectivity of the cingulo-

opercular network and the somato-cognitive action network also showed highest connectivity in the centromedian nucleus, highly overlapping with the peak connectivity of the target heat map (all three maps are thresholded at $R \geq 0.2$; slices taken at $z = 1$ and $y = 20$). a = anterior; p = posterior; s = superior; i = inferior.

p. 16: “Within the thalamus, the target heat map peaked in the anterior and superior part of the CM, suggesting that this region may represent the optimal stimulation target within the thalamus based on our analysis. However, this finding should be interpreted with caution due to the inherent limitations in spatial resolution associated with functional connectivity analyses. While the commonly used resolution of 2x2x2 mm in resting-state fMRI is adequate for seeding stimulation sites and investigating whole-brain functional networks, it may not be sufficient to precisely determine optimal target regions. An alternative approach would involve applying voxel-wise analyses of the electric field or the volume of activated tissue to generate probabilistic stimulation maps that predict therapeutic outcomes. Indeed, two previous studies employing such probabilistic stimulation maps found that targeting the anterior dorsal CM yielded optimal results, consistent with our findings^{27,61}. However, an earlier analysis failed to identify a clear pattern using such probabilistic maps, potentially due to the extended and heterogeneous follow-up periods of outcome assessments⁶². Therefore, further research with larger samples and well controlled outcome assessments are required to accurately delineate the optimal thalamic target for deep brain stimulation. A further important step could be to investigate whether similar functional networks as identified in the current analysis are critical for DBS of the globus pallidus internus.”

2. Although TS is typically described in a consistent clinical manner, it is well known that subtypes of tic prevalence or behavioral comorbidity can vary widely. Were there any consistent findings in network connectivity, based on preoperative resting-state or similar data, which correlated with specific tic subtypes or comorbidities? If so, were there any specific patterns that could predict or correlate with the best response to thalamic DBS?

Indeed, variability of tic subtypes and behavioral comorbidities in TS are significant and could potentially influence outcomes in DBS treatment. However, in our current study, we did not have access to preoperative physiological data across all centers, such as resting-state fMRI, that would allow us to correlate specific tic subtypes or comorbidities with network connectivity patterns.

To our knowledge, currently no clear preoperative clinical or physiological predictors of treatment response for DBS are known in GTS. Following the reviewer’s remark, we now additionally test whether the clinical data at hand would serve as predictors for treatment outcome. However, neither age, sex or the presence or absence of obsessive-compulsive behavior, as one of the most common comorbidity, was able to explain variance in outcomes (although age at surgery showed a weak, but not significant correlation).

In our study, we focused on the question whether the exact location and concomitant connectivity of stimulation sites would be able to explain outcome variance. However, we agree that the integration of other potential treatment predictors could significantly improve such models. Unfortunately, such clinical or physiological predictors are currently not at hand. We now discuss this aspect further in the limitations section. The following parts have been added to the manuscript:

p. 5: To investigate whether clinical and demographic factors could account for variations in treatment outcomes, we conducted a Spearman correlation analysis between patient age and the percentage change in YGTSS scores. Furthermore, we employed Wilcoxon rank-sum tests to compare treatment outcomes between female and male patients, as well as between patients with and without comorbid

obsessive-compulsive behavior. The presence of obsessive-compulsive behavior was assessed either through clinical transcripts or a minimum score of 7 on the Yale-Brown Obsessive Compulsive Scale³³.

p. 8: There was a weak but not-significant correlation between age and tic reduction ($r = -2.84$; $p = 0.088$) and no significant differences could be discerned between outcome of males and females ($z = 0.845$; $p = 0.423$) or outcomes of patients with and without OCB ($z = 0.258$; $p = 0.796$).

p. 18: The current analysis aimed to determine whether the connectivity of stimulation sites to specific functional networks could explain the variability in treatment outcomes. Ideally, this approach would be enhanced by incorporating additional clinical or biological predictors that could provide insights into the likelihood of responding to DBS surgery. Employing such predictors could also help refine the targeting of networks on a personalized basis. However, here and also, to the best of our knowledge, in the available literature to date no preoperative predictors could be identified for GTS.

3. If not, considering that functional imaging could data could well correlate with TS clinical presentation as it has been shown with tremor predominant or akinesia predominant PD, is there a point in using data from normal subjects to predict networks involved in pathological events and their neural correlates?

We agree that this aspect is crucial. As also requested by Reviewer 1, we have now expanded our discussion on this topic, as detailed below. Additionally, we want to highlight that this normative approach has been successfully used in the past to predict DBS outcomes across various conditions (for a review see: doi: 10.1016/j.neuroimage.2020.117180) and to explain the impacts of brain lesions on a wide range of symptoms (for a review see doi: 10.1097/WCO.0000000000001085). Nevertheless, incorporating disease-specific information about network connectivity could potentially enhance our models of optimal target networks, as now further elaborated in the limitations section:

p. 17f: “Further, the current study used a normative functional connectome that was derived from 1,000 healthy participants, thereby neglecting potential disease-specific connectome disruptions or individual connectivity perturbations. This approach serves as a broader characterization of the underlying stimulated or lesioned networks in an averaged, healthy brain, where the normative connectome can be understood as a generalized atlas of brain connectivity. This normative method has previously effectively predicted DBS treatment responses across various conditions^{38,39,44}. Beyond DBS, neural networks of numerous neuropsychiatric symptoms caused by heterogeneous focal brain lesions could be explained using the same fMRI data^{43,71–74}. Our investigation of lesions causing tic symptomatology matched the connectivity profile derived from DBS, indicating promising generalizability of the tic reduction map across brain interventions despite the usage of normative connectivity data. Nonetheless, patient- or disease-specific information about brain connectivity may allow to explain more of the variance in DBS outcomes. This is particularly of interest since GTS constitutes a neurodevelopmental disorder. Thus, it could be possible that the investigated networks develop differently across the lifespan in people with GTS. Indeed, an earlier study suggests that the CON shows altered maturation in adolescents with GTS compared to age-matched healthy controls⁷⁵. A similar conclusion, i.e. altered maturation of functional brain networks in GTS, was drawn from a more recent study. Here, a machine learning-derived classification of GTS using functional connectivity networks successfully distinguished patients with GTS from healthy controls— however, the distinguishing features differed between children and adults, pointing towards different neurodevelopmental trajectories of functional brain networks in children and adults with GTS⁷⁶. That being said, the principal functional brain networks were well traceable in the patient cohorts in both studies. Thus, although quantitative differences in these brain networks can be assumed, it appears unlikely that the general architecture of functional brain networks is drastically different in patients

with GTS. Along these lines, a study of patients undergoing DBS for Parkinson’s disease showed that the usage of patient-specific connectivity was not significantly superior to normative connectivity data and that both approaches revealed similar optimal brain networks to be targeted⁷⁷. Nonetheless, given the potential disease-specific brain alterations, we assume that at least some more variance of outcomes may be explainable by using disease-specific or even patient-specific connectivity data. However, obtaining high-quality resting-state fMRI data from unседated, severely affected patients with tic disorders that may qualify for DBS is hardly feasible, meaning that this approach is currently the only way to investigate functional connectivity estimates in patients with tic disorders who received DBS in a larger sample. One potential solution for this limitation could be to investigate a disease-specific functional group connectome derived from patients with less severe tics, where rs-fMRI of sufficient quality can be assessed. With such a group connectome, it could be tested whether predictability of outcomes could be improved compared to the currently used normative group connectome.”

4. Given that the insular region was highly involved in the positive correlation findings, could you provide higher resolution images focusing on the claustrum, a hypothetically involved network hub for consciousness that integrates associative cortical and basal ganglia networks? Could the claustrum also have been involved in the tic-producing network?

Indeed, the claustrum is a very interesting area for neuropsychiatric disorders. In the primary tic reduction network, derived from DBS sites, we now further added a visualization of the subcortical regions involved in this map. In this map, it can be seen that the insular cluster further encompasses parts of the posterior claustrum and of the posterior tip of the putamen. We added this description in the result section and provided a closer look at the subcortical contributions to the tic-reduction R-map as follows. The heat map and the lesion network map are not thresholded – in both maps, the claustrum is also positively associated with tic reduction. Since in both maps the goal was to compare the resulting maps with the predefined (cortical) action-related functional networks, we chose to focus on the visualization of the cortical patterns.

Insular and Subcortical Clusters of Tic-Reduction R-Map

p. 10: “The insular cluster encompassed the posterior short and long gyrus, as well as the middle and anterior short gyrus and the rolandic operculum. Subcortically, the insula cluster extended to posterior parts of the claustrum and encompassed the posterior tip of the putamen (Figure 3e).”

Finally, we need to correct an erroneous description of the statistical analysis that was used to compare the connectivity of tic-inducing lesions. When comparing connectivity values across networks, a Friedman test was employed to analyze paired samples across repeated measurements (i.e., networks). We have corrected the description in the manuscript (p. 9) and apologize for this oversight. Please note that the correct statistical procedure was properly implemented in the submitted code.

p. 8: "To this end, we performed a Friedman test between all connectivity values across networks for each lesion and a post-hoc comparison using the multcompare function in Matlab R2022b between CON and SCAN and all other networks (i.e. sensorimotor, central executive, visual, language and default mode network according to CAREN⁴² atlas) using an $p_{FDR} = 0.05$ to control for multiple comparisons."

We would like to thank the Reviewers for their assessment of our work. In the following, we will answer their comments and critiques point by point. We have numbered the comments by the reviewers for readability. The reviewer comments are displayed in green. Author answers are displayed in blue, with respective changes in the manuscript are displayed in red.

REVIEWER COMMENTS

Reviewer 1:

The authors have done an admirable job responding to my comments/suggestions.

We appreciate the reviewer's positive feedback.

Reviewer 2:

The authors have substantially improved the clarity of the manuscript and provided several new insights as a result of these revisions. I am satisfied that these changes meet the standard for plausibility and reproducibility. However, there are three wording changes that will improve the clarity of their message.

We thank the reviewer for the positive feedback and for the careful and thoughtful review.

You wrote "p. 8: "In the main analysis, we included 37 patients with severe GTS (mean age: 32.4 years \pm 10.8 standard deviations; range: 18 to 65; 5 females)." You are writing this incorrectly, and the solution was not to add the range of values. Your standard deviation = 10.8 years. It is NOT 10.8 standard deviations. Instead, write "32.4 years \pm 10.8 years (standard deviation); "

We thank the reviewer for clarifying our misunderstanding, we now adjusted the sentence accordingly:

p. 5: "In the main analysis, we included 37 patients with severe GTS (mean age: 32.4 years \pm 10.8 years (standard deviation); range: 18 to 65; 5 females)."

You wrote "p. 10: "... executive central 6% and 0.57; ..." Earlier in the paragraph you identify this network as central executive, but reverse it here.

We now changed the wording to "central executive".

You wrote "p 14: "... with fewer tics involving the feet and body core⁴⁹. Similar, the premonitory urge to tic..." You mean "Similarly" rather than "Similar".

We corrected this typo and now write "Similarly".